# GP-STPCA: Generalized Power Method for Sparse Tensor Principal Component Analysis

## Abstract

Sparse tensor principal component analysis (STPCA) seeks interpretable low-dimensional representations of high-order data by enforcing sparsity across tensor modes. However, the resulting optimization is highly nonconvex and computationally demanding, particularly in high-dimensional and unbalanced settings. We introduce GP-STPCA, a unified framework that reformulates STPCA into structured sparse PCA subproblems solvable via the generalized power method. Our approach accommodates both $\ell_0$- and $\ell_1$-penalties, in single-unit and block formulations, enabling efficient extraction of multiple sparse components. We provide theoretical guarantees by proving equivalence with the original sparse objective and analyzing convergence. Algorithmically, GP-STPCA further leverages efficient pattern-finding and post-processing to shrink the search space in column-dominant settings. Extensive experiments on synthetic recovery tasks, ImageNet reconstruction, and brain connectome analysis demonstrate that GP-STPCA consistently outperforms the SOTA sparseGeoHOPCA in terms of accuracy, sparsity control, interpretability, and computational efficiency.

## 1 Introduction

In this paper, we study the sparse tensor principal component analysis (STPCA) problem. Tensor PCA (TPCA) extends classical PCA to tensor-structured data for dimensionality reduction and pattern discovery (Kolda & Bader, 2009; Lu et al., 2008). In high-dimensional settings, sparsity plays a crucial role: it enhances interpretability, enables feature selection, improves statistical stability, and preserves tensor structure for applications such as multimodal learning (Sun et al., 2022), biomedical analysis (Allen, 2012), and recommender systems (Frolov & Oseledets, 2017).

However, introducing sparsity makes TPCA a non-convex and generally NP-hard problem (Hillar & Lim, 2013). Approximate algorithms have therefore been proposed: early work on multilinear PCA (Lu et al., 2006) laid the foundation for tensor analysis, while sparse HOSVD and sparse CP (Allen, 2012) incorporated sparsity-inducing penalties to recover interpretable low-rank structures. More recently, Xu et al. (2025) proposed sparseGeoHOPCA, a geometry-inspired framework that reformulates sparse higher-order PCA into binary optimization problems, improving both interpretability and efficiency. Further developments such as multilinear sparse PCA (Lai et al., 2014) have demonstrated effectiveness in image and video analysis (Liu et al., 2018), brain signal processing (Zhang et al., 2019), and biomedical data interpretation (Zhou et al., 2016). In contrast, within the conventional STPCA formulation, the resulting subproblems often involve far more columns than rows, a special case of sparse PCA that requires tailored algorithms for efficiency and accuracy.

We propose GP-STPCA, a unified and efficient method for solving the STPCA problem. For the matrix subproblems, we formulate four sparse PCA models under $\ell_0$- and $\ell_1$-constraints, in both single-unit and block forms, enabling the extraction of either a single dominant component or multiple components simultaneously.

As shown in Figure 1, the framework proceeds in three stages: (i) *Tensor preparation*: unfolding the input tensor along each mode to form sparse matrix PCA subproblems; (ii) *Sparse PCA subproblem*: applying the generalized power method to identify sparse patterns with convergence guarantees quickly; (iii) *Solution construction*: compressing the data via the identified patterns and assembling factor matrices with the core tensor to obtain a sparse multilinear decomposition. The method reduces the search space in column-dominant subproblems and ensures theoretical convergence.

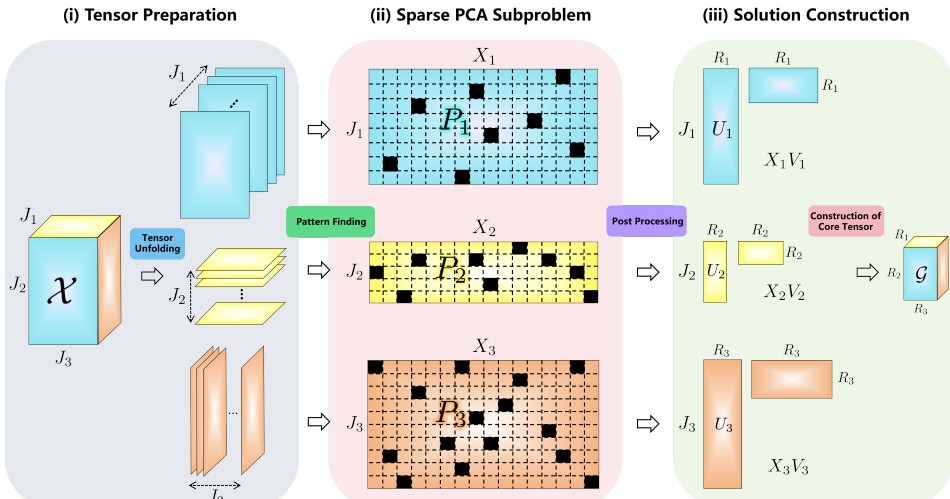

Figure 1: Illustration of the proposed GP-STPCA workflow on a third-order tensor. (i) The tensor $\mathcal{X}$ is unfolded mode by mode into matrices $X_1, X_2, X_3$. (ii) Sparse PCA subproblems are solved on each unfolding to identify sparsity patterns $P_1, P_2, P_3$. (iii) Based on these patterns, factor matrices $U_1, U_2, U_3$ and the core tensor $\mathcal{G}$ are constructed to yield a sparse multilinear decomposition.

**Main Contributions.** The key contributions of this work are summarized as follows:

- **Unified modeling:** We propose GP-STPCA, a generalized power method that reformulates sparse tensor PCA into a sequence of sparse matrix PCA subproblems, handling both $\ell_0$- and $\ell_1$-constraints penalties and single-unit or block formulations.

- **Dimensionality reduction and convex reformulation:** We reformulate each mode-wise optimization as a convex maximization problem over the Stiefel manifold, allowing sparse right factors to be identified first and orthogonal left factors obtained via SVD, thereby reducing the search dimension substantially.

- **Algorithmic framework:** We develop the overall STPCA algorithm together with efficient *pattern-finding* and *post-processing* schemes based on generalized power iterations, which identify sparse supports and maximize explained variance on the selected patterns.

- **Convergence properties:** We analyze the method as a generalized gradient scheme for convex maximization, and establish step-size convergence guarantees under strong convexity of either the function or the feasible set.

- **Empirical validation:** Through extensive experiments on synthetic support recovery, large-scale ImageNet image reconstruction, and brain connectome analysis, GP-STPCA consistently outperforms baselines in terms of accuracy, runtime efficiency, and interpretability.

## 2 PRELIMINARIES

We summarize the notations and review key results on sparse PCA and Tucker-based tensor PCA, which form the basis of our framework.

### 2.1 NOTATIONS AND DEFINITIONS

Unless stated otherwise, we adopt the following notation: scalars are denoted by lowercase letters (e.g., $a, b$), vectors by bold lowercase letters (e.g., $\mathbf{v}$), matrices by uppercase letters (e.g., $M$), and tensors by calligraphic letters (e.g., $\mathcal{T}$).

Let $\mathcal{X} \in \mathbb{R}^{J_1 \times J_2 \times \cdots \times J_N}$ be an $N$-th order tensor. Its mode-$n$ matricization, denoted by $X_{(n)} \in \mathbb{R}^{J_n \times \prod_{i \neq n} J_i}$, rearranges the mode-$n$ fibers of $\mathcal{X}$ into columns via the unfolding operator $\mathrm{unfold}_n(\mathcal{X})$. The inverse operation $\mathrm{fold}_n(\cdot)$ reconstructs the tensor from its matricized

form, satisfying $\mathcal{X} = \mathrm{fold}_n(X_{(n)})$. The mode-$k$ product (also known as the Tucker product) of $\mathcal{X}$ with a matrix $U_k \in \mathbb{R}^{J_k \times R_k}$ is denoted by $\mathcal{Y} = \mathcal{X} \times_k U_k$, and produces a tensor of size $\mathbb{R}^{J_1 \times \cdots \times J_{k-1} \times R_k \times J_{k+1} \times \cdots \times J_N}$. This transformation projects the mode-$k$ fibers of $\mathcal{X}$ onto a lower-dimensional subspace, and its matrix representation is given by: $Y_{(k)} = U_k X_{(k)}$, where $Y_{(k)}$ is the mode-$k$ matricization of the resulting tensor $\mathcal{Y}$.

The Stiefel manifold is the set of $n \times m$ matrices with orthonormal columns, $S_m^n = \{X \in \mathbb{R}^{n \times m} \mid X^\top X = I_m\}$. For $t \in \mathbb{R}$, we denote $\mathrm{sign}(t)$ as its sign and $t_+ = \max\{0, t\}$. The operator $\mathrm{Polar}(X)$ denotes the orthogonal factor in the polar decomposition of $X$.

## 2.2 Sparse Principal Component Analysis via the Generalized Power Method

A widely used formulation of sparse PCA introduces a sparsity-inducing penalty to promote sparsity in the loading vector. Given a data matrix $X \in \mathbb{R}^{n \times p}$, the single-unit problem is written as

$$\hat{w} = \arg \max_{\|w\|_2 = 1} \|Xw\|_2^2 - \lambda \|w\|_\zeta, \tag{1}$$

where $\lambda > 0$ controls the sparsity level and $\|\cdot\|_\zeta$ denotes a general sparsity measure. Here $\zeta = 1$ gives the lasso penalty, while $\zeta = 0$ corresponds to the cardinality penalty. To address Problem (1), Journée et al. (2010) proposed the generalized power method, a gradient-based scheme specifically designed for sparse PCA. This method reformulates the optimization as $\hat{z} = \arg \max_{\|z\|_2 = 1} \|S(X^\top z, \lambda)\|^2$, where the soft-thresholding operator $S(X^\top z, \lambda)$ is applied entry-wise. After obtaining $\hat{z}$, the sparse loading vector is recovered as $\hat{w} = S(X^\top \hat{z}, \lambda)/\|S(X^\top \hat{z}, \lambda)\|$, so that the nonzero support of $\hat{w}$ is directly determined by the thresholded pattern. The component weights are then obtained by applying PCA to the reduced matrix after discarding zeroed variables in $\hat{w}$. Thus, the original $p$-dimensional problem is reformulated as an $n$-dimensional one, which is advantageous when $p \gg n$.

## 2.3 Tucker-Based Tensor Principal Component Analysis

Tucker-based tensor PCA generalizes classical PCA to higher-order data by exploiting the Tucker decomposition. For an $N$th-order tensor $\mathcal{X} \in \mathbb{R}^{J_1 \times \cdots \times J_N}$, the model is

$$\mathcal{X} \approx \mathcal{G} \times_1 U_1 \times_2 U_2 \cdots \times_N U_N, \tag{2}$$

where $\mathcal{G} \in \mathbb{R}^{R_1 \times \cdots \times R_N}$ is the core tensor and $U_n \in \mathbb{R}^{J_n \times R_n}$ are orthonormal factor matrices. In element-wise form: $\mathcal{X}(i_1, \ldots, i_N) \approx \sum_{\alpha_1=1}^{R_1} \cdots \sum_{\alpha_N=1}^{R_N} \mathcal{G}(\alpha_1, \ldots, \alpha_N) \prod_{n=1}^{N} U_n(i_n, \alpha_n)$. This multilinear form reduces storage from $\mathcal{O}(J_1 \cdots J_N)$ to $\mathcal{O}(R_1 \cdots R_N + \sum_{n=1}^{N} J_n R_n)$, and methods such as higher-order SVD or alternating least squares are commonly used to compute the resulting interpretable components.

# 3 Tucker-Based Sparse Tensor Principal Component Analysis

In this section, we develop a Tucker-based framework for sparse tensor principal component analysis (STPCA). By unfolding the tensor along each mode, the problem is transformed into sparse matrix PCA subproblems with different sparsity-inducing formulations. Building upon this reformulation, we introduce the proposed GP-STPCA algorithm and establish its theoretical foundations.

## 3.1 Problem Formulation

To enhance interpretability and robustness in multilinear data analysis, Tucker-based tensor PCA can be extended with sparsity constraints, leading to the Tucker-based sparse tensor PCA (STPCA). Let $\mathcal{X} \in \mathbb{R}^{J_1 \times \cdots \times J_N}$ be the data tensor and $U_n \in \mathbb{R}^{J_n \times R_n}$ the projection matrix for each mode. The objective is to minimize the projection error while enforcing sparsity and orthogonality, so the STPCA problem is thus formalized as:

$$\begin{aligned} \underset{U_1, \ldots, U_N}{\text{minimize}} \quad & \|\mathcal{X} - \mathcal{X} \times_1 U_1 U_1^\top \cdots \times_N U_N U_N^\top\|_F^2 \\ \text{subject to} \quad & \|\mathcal{X} \times_1 U_1 \cdots \times_N U_N\|_\zeta \leq k, \quad U_n \in S_{R_n}^{J_n}, \quad \text{for } n = 1, \ldots, N, \end{aligned} \tag{3}$$

where $k > 0$ control the sparsity of core tensor and $\| \cdot \|_\zeta$ denotes a sparsity measure, with $\zeta = 1$ (lasso) or $\zeta = 0$ (cardinality). This formulation yields low-dimensional multilinear projections with sparse and interpretable factors across modes. The STPCA problem is inherently challenging due to its non-convex nature and lack of a closed-form solution.

## 3.2 Decomposition into mode-wise subproblems

Inspired by the alternating optimization strategy in Tucker decomposition, we decompose the problem into $N$ independent subproblems.

**Theorem 3.1** *Let* $(U_1, \ldots, U_{n-1}, U_{n+1}, \ldots, U_N)$ *be fixed. Then the optimization of* $U_n$ *in (3) reduces to the sparse matrix approximation problem*

$$\min_{U_n} \quad \|X_n - U_n U_n^\top X_n\|_F^2 \quad s.t. \ \|\mathcal{X} \times_{j \neq n} U_j\|_\zeta \leq k_n, \ \ U_n \in S_{R_n}^{J_n}. \tag{4}$$

*where* $X_n = \mathrm{unfold}_n(\mathcal{X} \times_{j \neq n} U_j U_j^\top)$, *and* $k_n$ *denotes the sparsity level adapted to mode* $n$.

*Proof.* See Appendix A. $\qquad\qquad\qquad\qquad\qquad\qquad\qquad\qquad\qquad\qquad\qquad\square$

**Remark 3.1** *Note that the conclusion in (4) is derived under the setting where the factor matrix* $U_n$ *is updated iteratively. If* $U_n$ *is instead generated in a single step, or when no initialization is provided, then* $X_n$ *should be directly taken as* $X_{(n)}$.

## 3.3 Overall STPCA Framework

Based on the preceding analysis, we now present the basic algorithmic framework for solving STPCA, as summarized in Algorithm 1 (See Appendix B).

## 3.4 Reformulation as Sparse PCA Subproblem

To solve this subproblem, we begin with the mode-$n$ subproblem formulated as (4). Since $U_n U_n^\top$ is an orthogonal projector, the Pythagorean identity gives $\|X_n\|_F^2 = \|X_n - U_n U_n^\top X_n\|_F^2 + \|U_n^\top X_n\|_F^2$, hence (4) is equivalent to

$$\max_{U_n} \quad \|U_n^\top X_n\|_F^2 \quad \text{s.t. } \|\mathcal{X} \times_{j \neq n} U_j\|_\zeta \leq k_n, \ \ U_n \in S_{R_n}^{J_n}. \tag{5}$$

Since $X_n$ has far more columns than rows, it is natural to seek a right sparse factor $V_n$; computing the SVD of the compressed matrix $X_n V_n$ then yields $U_n$. Thus, the problem reduces to finding a sparse right factorization of $X_n$. For clarity, we focus on the case $\zeta = 1$, while the case $\zeta = 0$ is deferred to Appendix C. Following the generalized power method introduced in Section 2.2, we adopt two strategies for multiple sparse components: (i) the single-unit approach with sequential deflation, and (ii) the block sparse approach computing multiple components jointly.

## 3.5 Sparse PCA formulations via $\ell_1$ penalty

We first consider the $\ell_1$-penalized setting, which promotes sparsity through soft thresholding.

### 3.5.1 Single-unit sparse PCA via $\ell_1$ penalty

We first consider the single-unit case, where one sparse vector is extracted at a time. The $\ell_1$-penalized formulation reads

$$\phi_{\ell_1}^n(\gamma^n) \stackrel{\text{def}}{=} \max_{v_n^\top v_n \leq 1} \|X_n v_n\|_2 - \gamma^n \|v_n\|_1. \tag{6}$$

Noticing that $\|X_n v_n\|_2 = \max_{z \in S_1^{J_n}} z^\top X_n v_n$, we can reformulate the problem as

$$\phi_{\ell_1}^n(\gamma^n) = \max_{z \in S_1^{J_n}} \max_{v_n} z^\top X_n v_n - \gamma^n \|v_n\|_1. \tag{7}$$

For fixed $z$, the inner maximization over $v_n$ admits a closed-form solution:

$$v_n(i)^*(\gamma^n) = \frac{\text{sign}(X_n(:,i)^\top z)\,[\,|X_n(:,i)^\top z| - \gamma^n\,]_+}{\sqrt{\sum_{k=1}^{\prod_{p \neq n} J_p}[\,|X_n(:,k)^\top z| - \gamma^n\,]_+^2}}, \quad i = 1, \ldots, \prod_{k \neq n} J_k. \tag{8}$$

Substituting this expression back yields a reformulated objective:

$$\boxed{\phi_{\ell_1}^{n}{}^2(\gamma^n) = \max_{z \in S_1^{J_n}} \sum_{i=1}^{\prod_{k \neq n} J_k} [\,|X_n(:,i)^\top z| - \gamma^n\,]_+^2,} \tag{9}$$

which is a smooth objective defined on the Stiefel manifold $S_1^{J_n}$ and is more efficient to optimize than the original formulation.

**Deflation Scheme** To obtain multiple sparse components from the single-unit method, we adopt the classical deflation strategy (d'Aspremont et al., 2007). Given a unit-norm sparse vector $v \in \mathbb{R}^n$ of $X \in \mathbb{R}^{p \times n}$, let $z = Xv$ be the associated score that solves $\min_{z \in \mathbb{R}^p} \|X - zv^\top\|_F$. Subsequent directions are extracted from the residual $X - zv^\top$, thereby removing explained variance and enforcing complementarity. Refined deflation variants (Mackey, 2008) further enhance stability and orthogonality.

### 3.5.2 BLOCK SPARSE PCA VIA $\ell_1$ PENALTY

The block formulation extends the single-unit case to extract multiple components simultaneously. Using Lagrange multipliers, Problem (4) can be reformulated as the $\ell_1$-penalized block problem

$$\psi_{\ell_1,R_n}^n(\gamma^n) \stackrel{\text{def}}{=} \max_{Z \in S_{R_n}^{J_n}, \text{diag}(V_n^\top V_n) = I_{R_n}} \text{Tr}(Z^\top X_n V_n N^n) - \sum_{j=1}^{R_n} \gamma_j^n \sum_{i=1}^{\prod_{k \neq n} J_k} |V_n(i,j)|, \tag{10}$$

where $\gamma^n = [\gamma_1^n, \ldots, \gamma_{R_n}^n]^\top \geq 0$ and $N^n = \text{diag}(\mu_1^n, \ldots, \mu_{R_n}^n)$ with positive entries, representing relative weights associated with different principal components. Since the columns of $V_n$ are independent, Problem (10) decouples as

$$\psi_{\ell_1,R_n}^n(\gamma^n) = \max_{Z \in S_{R_n}^{J_n}} \sum_{j=1}^{R_n} \max_{\|V_n(:,j)\|_2 = 1} \mu_j^n Z(:,j)^\top X_n V_n(:,j) - \gamma_j^n \|V_n(:,j)\|_1. \tag{11}$$

The optimal $V_n$ columns admit the closed form

$$V_n(i,j)^* = \frac{\text{sign}(X_n(:,i)^\top Z(:,j))\,[\,\mu_j^n |X_n(:,i)^\top Z(:,j)| - \gamma_j^n\,]_+}{\sqrt{\sum_{k=1}^{\prod_{p \neq n} J_p}[\,\mu_j^n |X_n(:,k)^\top Z(:,j)| - \gamma_j^n\,]_+^2}}. \tag{12}$$

Substituting back yields the equivalent formulation

$$\boxed{\psi_{\ell_1,R_n}^{n}{}^2(\gamma^n) = \max_{Z \in S_{R_n}^{J_n}} \sum_{j=1}^{R_n} \sum_{i=1}^{\prod_{k \neq n} J_k} [\,\mu_j^n |X_n(:,i)^\top Z(:,j)| - \gamma_j^n\,]_+^2,} \tag{13}$$

which maximizes a convex function $f : \mathbb{R}^{J_n \times R_n} \to \mathbb{R}$ on on the Stiefel manifold $S_{R_n}^{J_n}$. Both the single-unit (9) and block (13) formulations (as well as their $\ell_0$ counterparts) fall within a unified optimization framework. They can be efficiently solved using the generalized power method, which applies a gradient-based scheme to maximize convex functions over compact feasible sets. For completeness, we next describe this scheme and its stepsize convergence properties.

### 3.6 STEPSIZE CONVERGENCE OF THE GRADIENT SCHEME

We analyze the convergence of the generalized gradient scheme for sparse PCA subproblems. Let $f : E \to \mathbb{R}$ be a convex function on a finite-dimensional space $E$, and consider

$$f^* = \max_{x \in Q} f(x), \tag{14}$$

with $Q \subseteq E$ compact. Here $f'(x)$ denotes any subgradient and $\partial f(x)$ its subdifferential. We solve this problem using Algorithm 2 (See Appendix D), which can be viewed as a generalized power method cast as a gradient scheme for convex maximization. Convergence requires mild structural conditions: either strong convexity of $f$ or strong convexity of $Conv(Q)$.

**Assumption 1** *The subgradient norms of $f$ are uniformly bounded away from zero on $Q$, which means $\delta_f = \min_{x \in Q, f'(x) \in \partial f(x)} \|f'(x)\|_* > 0$, where $\|\cdot\|_*$ is the dual norm.*

**Assumption 2** *$f$ is strongly convex: there exists $\sigma_f > 0$ such that for all $x, y \in E$, $f(y) \geq f(x) + \langle f'(x), y - x \rangle + \frac{\sigma_f}{2} \|y - x\|^2$.*

**Assumption 3** *The convex hull $Conv(Q)$ is strongly convex, i.e., for any $x, y \in Conv(Q)$ and $\alpha \in [0, 1]$, $\alpha x + (1 - \alpha)y + \frac{\sigma_Q}{2}\alpha\|x - y\|^2 S \subset Conv(Q)$, where $S$ is the unit ball in $E$.*

**Theorem 3.2** *(Stepsize Convergence) Let $f$ be convex, and let either Assumption 2 or Assumption 1 and 3 be satisfied. If $\{x_k\}_{k \geq 0}$ is the sequence of points generated by the Algorithm 2, then*

$$\sum_{k=0}^{\infty} \|x_{k+1} - x_k\|^2 \leq \frac{2(f^* - f(x_0))}{\sigma_Q \delta_f + \sigma_f}. \tag{15}$$

*Proof.* See Appendix E. $\qquad\qquad\square$

**Remark 3.2** *Theorem 3.2 shows that for the gradient method scheme in Algorithm 2, in order to produce an iterate satisfying $\min_{0 \leq i \leq k} \|x_{i+1} - x_i\| \leq \epsilon$, we require at most $k = O(\epsilon^{-2})$ iterations. This iteration bound aligns with the global complexity of standard first-order methods.*

### 3.7 Algorithms for Sparse PCA Subproblem with $\ell_1$ Penalty

The sparse PCA formulations in Section 3.5 produce locally optimal sparsity patterns, either in the single-unit or block setting. While penalty terms enforce sparsity, they may also distort the values of active entries. Thus, an effective algorithm consists of two stages: (*i*) identifying a sparsity pattern, and (*ii*) estimating the active entries to maximize explained variance. In the following, we focus on the $\ell_1$-penalized formulations, while the $\ell_0$ counterparts are deferred to Appendix G. We present the general block formulation here, which reduces to the single-unit case when $R_n = 1$.

#### 3.7.1 Pattern-finding

Applying the gradient scheme (Algorithm 2) to the optimization Problems (9), and (13), yields Algorithms 3 and 4 (both in Appendix F), which determine a binary sparsity mask $P \in \{0,1\}^{\prod_{i \neq n} J_i \times R_n}$. Here $P(i, j) = 1$ indicates that coefficient $V_n(i, j)$ is active, and $P(i, j) = 0$ otherwise. The per-iteration cost of the single-unit methods (Algorithms 3) is $O(\prod_{n=1}^{N} J_n)$, while the block methods (Algorithms 4) require $O(R_n \prod_{n=1}^{N} J_n)$ operations. Thresholds $\gamma^n$ are chosen below natural upper bounds: $\gamma^n \leq \max_i \|X_n(:, i)\|_2$ for $\ell_1$ single-unit, and $\gamma_j^n \leq \max_i \mu_j^n \|X_n(:, i)\|_2$ for $\ell_1$ block.

#### 3.7.2 Post-processing

With pattern $P$ fixed, the active entries of $V_n$ are refined to maximize variance. We solve

$$(Z^*, V_n^*) \stackrel{\text{def}}{=} \max_{Z \in S_{R_n}^{J_n}; \text{diag}(V_n^\top V_n) = I_{R_n}; V_n|_{P'} = 0} \text{Tr}(Z^\top X_n V_n N^n). \tag{16}$$

where $P'$ is the complement of $P$. In the single-unit case, the solution is the leading SVD of $X_n|_P$:

$$Z^* = u, \qquad V_n^*|_P = v, \qquad V_n^*|_{P'} = 0, \tag{17}$$

where $\sigma u v^\top$ is a rank-one decomposition of $X_n|_P$. This coincides with the variational renormalization in Moghaddam et al. (2005). In the block case, (16) is solved by alternating optimization:

**Theorem 3.3** *Consider the optimization problem*

$$\max_{Z,V_n} \mathrm{Tr}(Z^\top X_n V_n N^n), \; s.t. \; Z \in S_{R_n}^{J_n}; \mathrm{diag}(V_n^\top V_n) = I_{R_n}; V_n|_{P'} = 0. \tag{18}$$

*For a fixed $V_n$, the optimal $Z^*$ is given by the U factor of the polar decomposition of $X_n V_n N^n$. Conversely, for a fixed $Z \in S_{R_n}^{J_n}$, the optimal $V_n^*$ is characterized by $V_n^*|_P = (X_n^\top Z N^n D)|_P, \quad V_n^*|_{P'} = 0$ where D is a positive diagonal matrix that normalizes each column of $V_n^*$ to unit norm, that is, $D = \mathrm{diag}(N^n Z^\top X_n X_n^\top Z N^n)^{-\frac{1}{2}}$.*

*Proof.*    See Appendix H.    □

The alternating optimization scheme is summarized in Algorithm 7 (See Appendix I), initialized by an accumulation point of the pattern-finding step (Algorithms 4 and 6). This postprocessing heuristic is strictly required only for the $\ell_1$ block case. For $\ell_0$ formulations, since the penalty depends only on the sparsity pattern $P$ and not on the values of $V_n|_P$, the solutions of Algorithms 5 or 6 already serve as local maximizers of (16), providing a direct alternative to Algorithm 7. In the single-unit $\ell_1$ case (Algorithm 3), the solution (17) is available.

**Remark 3.3** *We have introduced all necessary technical tools. Here we summarize the four variants of GP-STPCA (single/block with $\ell_0$- or $\ell_1$-penalty) and describe the initialization of hyperparameters. Detailed settings can be found in Appendix J.*

## 4    EMPIRICAL RESULTS

This section reports synthetic and real-data experiments evaluating the proposed *GP-STPCA* framework. We first evaluate sparse support recovery in controlled simulations, then test image reconstruction on ImageNet, and finally analyze brain connectomes, where GP-STPCA consistently outperforms baselines in accuracy, efficiency, and interpretability. All experiments were run on a workstation with an Intel i7-10700 CPU (2.90GHz), NVIDIA RTX 4070 Super GPU, and 64GB RAM. Hyperparameter initialization is given in Remark 3.3.

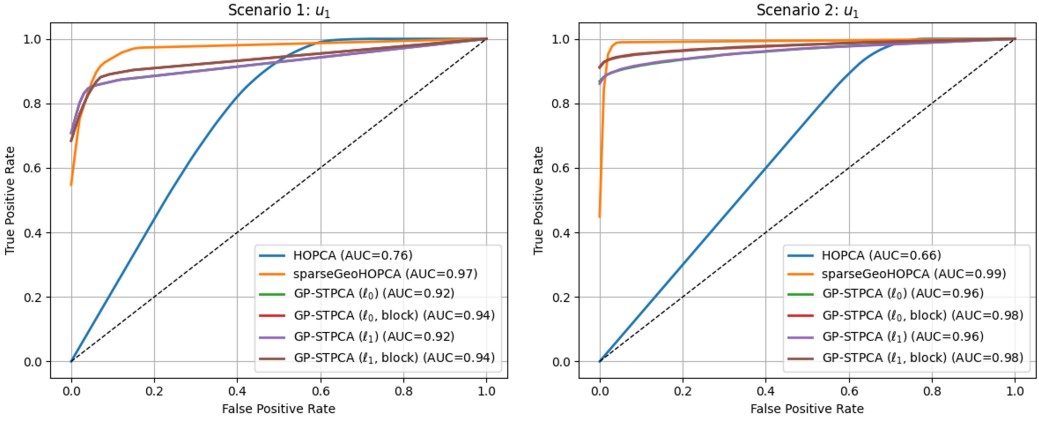

Figure 2: ROC curves for mode-$u_1$ in Scenarios 1 and 2, where it is the only sparse mode. Results are averaged over fifty replicates. Across both settings, GP-STPCA variants achieve consistently high true positive rates and large AUC values (up to 0.98), demonstrating accurate and stable support recovery. As further shown in the runtime comparison of Appendix K.3, GP-STPCA achieves 50–100× speedup over existing baselines.

### 4.1    SYNTHETIC EXPERIMENTS ON SPARSE SUPPORT RECOVERY

We evaluate the support recovery performance of four variants of *GP-STPCA* (different penalties and block choices) using synthetic low-rank third-order tensor models under varying sparsity and

dimensionality. The observed tensor is generated as

$$\mathcal{X} = \sum_{k=1}^{K} d_k \, \mathbf{u}_k \circ \mathbf{v}_k \circ \mathbf{w}_k + \mathcal{E}, \quad \mathcal{E}_{i,j,l} \overset{\text{iid}}{\sim} \mathcal{N}(0,1), \tag{19}$$

with $K = 1$ and $d_1 = 100$. We compare against both the classical HOPCA, to illustrate the benefit of incorporating sparsity, and the state-of-the-art sparseGeoHOPCA (Xu et al., 2025). Four scenarios with different tensor sizes and sparsity patterns are considered (details in Appendix K.1). Results show that the block variants of *GP-STPCA* achieve consistently better recovery, while differences between $\ell_0$ and $\ell_1$ penalties are minor. Overall recovery accuracy is comparable to sparseGeo-HOPCA, but our method is significantly faster. Figure 2 reports ROC curves averaged over 50 trials in both balanced and unbalanced settings (with sparsity only on $u_1$), demonstrating strong recovery performance.

Appendix K.2 presents ROC curves for Scenarios 3 and 4, and Appendix K.3 reports comparisons of true/false positive rates and computational efficiency among HOPCA, *sparseGeoHOPCA*, and the proposed GP-STPCA variants.

## 4.2 IMAGE RECONSTRUCTION

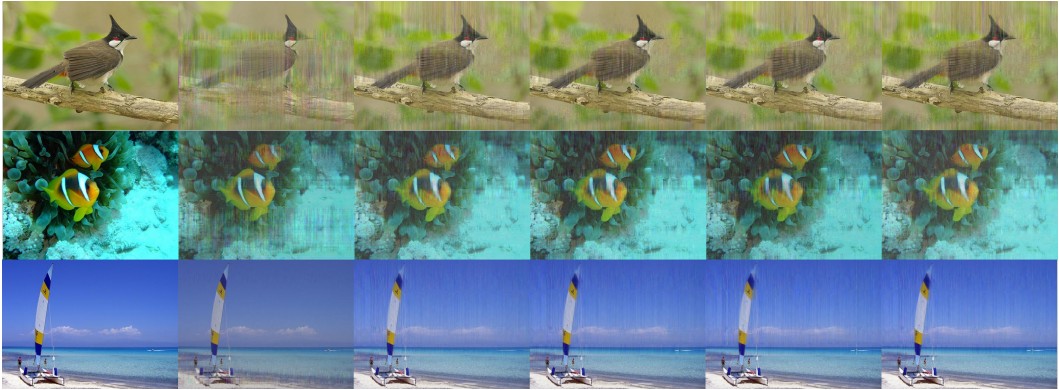

Figure 3: Visual comparison of ImageNet reconstructions for three representative examples (from top to bottom: Image1, Image2, and Image3). From left to right: original image, *sparseGeo-HOPCA*, GP-STPCA($\ell_0$), GP-STPCA($\ell_0$, block), GP-STPCA($\ell_1$), and GP-STPCA($\ell_1$, block). The GP-STPCA variants consistently enhance reconstruction quality: $\ell_0$-based methods effectively preserve salient object shapes, block extensions further improve color uniformity, and $\ell_1$-based methods achieve sharper edge recovery and finer texture details. Overall, GP-STPCA demonstrates superior balance between structural preservation and visual clarity across diverse images.

We further examine the effectiveness of the proposed *GP-STPCA* framework by testing its four variants (different penalties and block choices) on random ImageNet samples (Russakovsky et al., 2015). After extracting a fixed number of sparse components, the original images are reconstructed to evaluate visual fidelity and quantitative performance. Figure 3 presents representative results.

Table 1: PSNR (dB) comparison on Image1–3 in Figure 3.

| Method | Image1 | Image2 | Image3 |
|---|---|---|---|
| sparseGeoHOPCA | 21.491 | 20.965 | 27.720 |
| GP-STPCA ($\ell_0$) | 26.412 | 22.949 | 31.438 |
| GP-STPCA ($\ell_0$, block) | **27.692** | 22.243 | 30.765 |
| GP-STPCA ($\ell_1$) | 26.419 | 22.816 | 31.660 |
| GP-STPCA ($\ell_1$, block) | 26.263 | **23.054** | **31.727** |

Across all samples, GP-STPCA variants achieve sharper textures and fewer directional artifacts compared to the baseline sparseGeoHOPCA, which tends to produce blurrier reconstructions. Table 1 further confirms these observations: block-based variants consistently yield higher PSNR, with GP-STPCA ($\ell_1$, block) delivering the best overall reconstruction quality.

See Appendix L for preprocessing, settings, and additional results.

### 4.3 CONNECTOME-BASED ANALYSIS OF BRAIN NETWORK

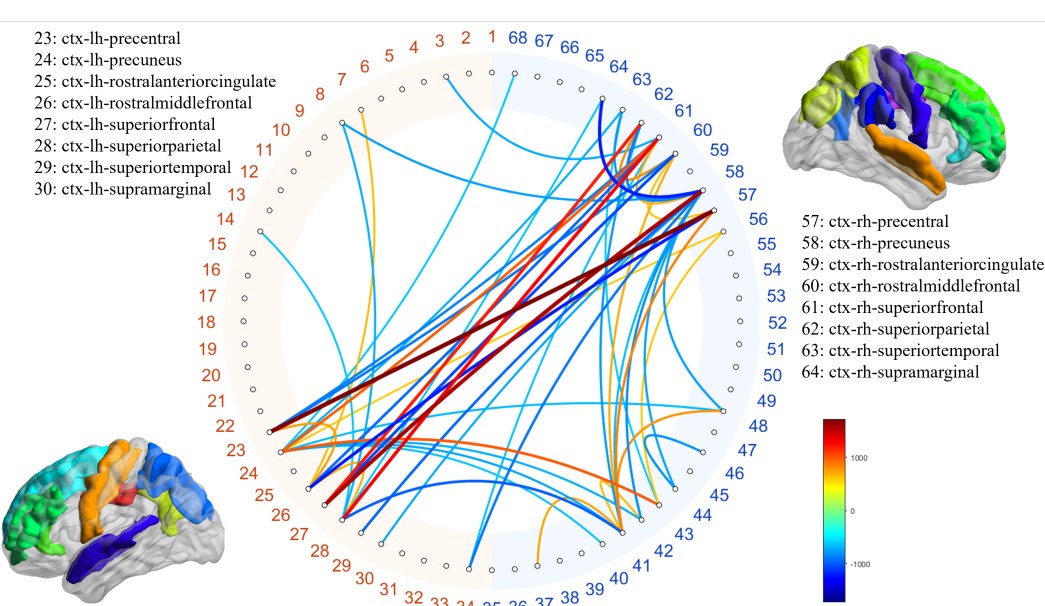

Figure 4: Chord diagram illustrating connectivity differences between high- and low-reading groups. Nodes are cortical regions (left hemisphere: indices 23–30; right hemisphere: indices 57–64), with labels from the Desikan–Killiany atlas. Colored chords represent inter- and intra-hemispheric connectivity differences, with line thickness denoting magnitude and color encoding direction (red/yellow vs. blue). Highlighted regions (precentral, precuneus, superiorfrontal, supramarginal) show consistent alterations in structural connectivity associated with reading performance.

As an application, we study the relationship between structural connectomes and the age-adjusted English reading score using HCP data (Van Essen et al., 2013). Structural connectomes were derived from 200 extreme subjects of the Human Connectome Project, yielding $68 \times 68 \times 200$ adjacency tensors based on the Desikan–Killiany atlas (Desikan et al., 2006) parcellation. GP-STPCA extracts discriminative cross-hemispheric fronto-parietal connections, indicating that higher reading ability is associated with stronger inter-hemispheric integration. Figure 4 visualizes the top-50 discriminative edges, where warm colors (red–yellow) highlight stronger connections in the high-reading group and cool colors (blue) indicate stronger connections in the low-reading group. Details on the experiments are provided in Appendix M.

## 5 CONCLUSION

In this work, we proposed GP-STPCA, a unified framework for sparse tensor principal component analysis. By reformulating the original nonconvex problem into structured sparse matrix PCA subproblems and solving them via the generalized power method, GP-STPCA accommodates both $\ell_0$- and $\ell_1$-penalties under single-unit and block formulations. Our framework offers theoretical guarantees through equivalence with the original sparse objective and convergence analysis, while algorithmically exploiting pattern-finding and post-processing to reduce the search space in column-dominant settings.

Extensive experiments on synthetic recovery, large-scale ImageNet reconstruction, and brain connectome analysis demonstrate clear advantages in accuracy, sparsity control, interpretability, and computational efficiency over existing approaches such as sparseGeoHOPCA. These results highlight the potential of GP-STPCA as a versatile and scalable tool for high-dimensional tensor data analysis.

## REPRODUCIBILITY STATEMENT

The Python implementation of the proposed method is available in the supplementary materials.

## ETHICS STATEMENT

This work does not involve human subjects, personally identifiable data, or sensitive information. All datasets used are publicly available and widely adopted in the research community. The proposed methods focus on algorithmic development and empirical validation. We believe this work raises no ethical concerns in relation to the ICLR Code of Ethics.

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

## A PROOF OF THEOREM 3.1

This appendix provides the detailed proof of Theorem 3.1.

*Proof.* Let $\{U_i\}_{i=1,\,i\neq n}^{N}$ denote the collection of all mode-$i$ projection matrices excluding mode $n$, i.e., $(U_1, \ldots, U_{n-1}, U_{n+1}, \ldots, U_N)$. We denote the objective function in (3) as $f(U_n)$ when all other $U_i$'s are fixed. Assume that each $U_i$ is column-orthonormal.

By properties of mode-wise tensor projections and the orthogonality of $U_i U_i^\top$, we have:

$$
\begin{aligned}
f(U_n) &= \left\| \mathcal{X} - \mathcal{X} \times_1 U_1 U_1^\top \cdots \times_N U_N U_N^\top \right\|_F^2 \\
&= \left\| \mathcal{X} \times_N U_N U_N^\top - \mathcal{X} \times_1 U_1 U_1^\top \cdots \times_N U_N U_N^\top \right\|_F^2 \\
&\quad + \left\| \mathcal{X} - \mathcal{X} \times_N U_N U_N^\top \right\|_F^2 \\
&\;\;\vdots \\
&= \left\| \mathcal{X} \times_{j\neq n} U_j U_j^\top - \mathcal{X} \times_{j\neq n} U_j U_j^\top \times_n U_n U_n^\top \right\|_F^2 \\
&\quad + \left\| \mathcal{X} - \mathcal{X} \times_{j\neq n} U_j U_j^\top \right\|_F^2 ,
\end{aligned}
\tag{20}
$$

where $\times_{j\neq n}$ denotes the sequence of mode-$j$ projections over all $j \in \{1, \ldots, N\} \setminus \{n\}$. Noting that the second term in the final expression is independent of $U_n$, the optimization reduces to minimizing $\left\| \mathcal{X} \times_{j\neq n} U_j^\top - \mathcal{X} \times_{j\neq n} U_j^\top \times_n U_n U_n^\top \right\|_F^2$. Let $X_n = \mathrm{unfold}_n(\mathcal{X} \times_{j\neq n} U_j U_j^\top)$ and this is equivalent to minimizing the matrix-form objective $\left\| X_n - U_n U_n^\top X_n \right\|_F^2$, which corresponds precisely to Problem (3) in the main text, where $k_n$ denotes the sparsity level adapted to mode $n$. This completes the proof. □

## B ALGORITHM FOR STPCA FRAMEWORK

---

**Algorithm 1:** STPCA framework

---

**Input:** Tensor $\mathcal{X} \in \mathbb{R}^{J_1 \times J_2 \times \cdots \times J_N}$; target ranks $(R_1, \ldots, R_N)$; sparsity levels $\{k_n\}_{n=1}^N$.
**Output:** Factor matrices $\{U_n\}_{n=1}^N$ and core tensor $\mathcal{G}$.

1 **for** $n = 1$ *to* $N$ **do**

2 $\quad$ Compute $X_n = X_{(n)} \in \mathbb{R}^{J_n \times \prod_{i\neq n} J_i}$;

3 $\quad$ Solve mode-$n$ sparse matrix subproblem:

$$
\min_{U_n} \; \|X_n - U_n U_n^\top X_n\|_F^2 \quad \text{s.t. } \|\mathcal{X} \times_{j\neq n} U_j\|_\zeta \leq k_n, \; U_n \in S_{R_n}^{J_n}.
$$

4 $\quad$ Set $U_n \in \mathbb{R}^{J_n \times R_n}$ as the final solution for mode $n$;

5 Compute the core tensor as: $\mathcal{G} = \mathcal{X} \times_1 U_1^\top \times_2 U_2^\top \cdots \times_N U_N^\top$.

---

## C SPARSE PCA FORMULATIONS VIA $\ell_0$ PENALTY

This appendix provides the sparse PCA formulations via $\ell_0$-penalty.

### C.1 SINGLE-UNIT SPARSE PCA VIA $\ell_0$ PENALTY

We consider the cardinality-penalized formulation, which directly enforces sparsity by penalizing the number of nonzero loadings. The single-unit case is given by

$$
\phi_{\ell_0}^n(\gamma^n) \stackrel{\text{def}}{=} \max_{v_n^\top v_n \leq 1} \|X_n v_n\|_2^2 - \gamma^n \|v_n\|_0.
\tag{21}
$$

Since $\|X_n v_n\|_2^2 = \max_{z \in S_1^{J_n}} (z^\top X_n v_n)^2$, the problem can be rewritten as

$$
\phi_{\ell_0}^n(\gamma^n) = \max_{z \in S_1^{J_n}} \max_{v_n} (z^\top X_n v_n)^2 - \gamma^n \|v_n\|_0.
\tag{22}
$$

For fixed $z$, the inner maximization has the explicit solution

$$v_n(i)^*(\gamma^n) = \frac{[\text{sign}((X_n(:,i)^\top z)^2 - \gamma^n)]_+ X_n(:,i)^\top z}{\sqrt{\sum_{k=1}^{\prod_{p \neq n} J_p} [\text{sign}((X_n(:,i)^\top z)^2 - \gamma^n)]_+ (X_n(:,i)^\top z)^2}}, \quad i = 1, \ldots, \prod_{k \neq n} J_k. \tag{23}$$

Substituting back yields the simplified objective

$$\phi_{\ell_0}^n(\gamma^n) = \max_{z \in S_1^{J_n}} \sum_{i=1}^{\prod_{k \neq n} J_k} [(X_n(:,i)^\top z)^2 - \gamma^n]_+, \tag{24}$$

which defines a smooth problem on the Stiefel manifold $S_1^{J_n}$, more tractable than the original non-convex form.

**Deflation Scheme** As in the $\ell_1$ case, multiple components can be obtained by deflation (d'Aspremont et al., 2007). Given a sparse loading $v$, its score $z = Xv$ that solves $\min_{z \in \mathbb{R}^p} \|X - zv^\top\|_F$ is computed, and subsequent directions are extracted from the residual $X - zv^\top$. This sequential removal of explained variance ensures complementary components. Refined deflation methods (Mackey, 2008) improve stability and orthogonality.

## C.2 BLOCK SPARSE PCA VIA $\ell_0$ PENALTY

Extending to the block setting with multiple components, the $\ell_0$-penalized formulation reads

$$\psi_{\ell_0, R_n}^n(\gamma^n) \overset{\text{def}}{=} \max_{Z \in S_{R_n}^{J_n}, \text{diag}(V_n^\top V_n) = I_{R_n}} \text{Tr}\big(\text{diag}(Z^\top X_n V_n N^n)^2\big) - \sum_{j=1}^{R_n} \gamma_j^n \|V_n(:,j)\|_0. \tag{25}$$

where $\gamma^n = [\gamma_1^n, \ldots, \gamma_{R_n}^n]^\top \geq 0$ controls sparsity and $N^n = \text{diag}(\mu_1^n, \ldots, \mu_{R_n}^n)$ is positive diagonal, representing relative weights associated with different principal components. Since the problem decouples across columns of $V_n$, we obtain

$$\psi_{l_0, R_n}^n(\gamma^n) = \max_{Z \in S_{R_n}^{J_n}} \sum_{j=1}^{R_n} \max_{\|V_n(:,j)\|_2 = 1} (\mu_j^n Z(:,j)^\top X_n V_n(:,j))^2 - \gamma_j^n \|V_n(:,j)\|_0 \tag{26}$$

Thus, each column $V_n(:,j)$ admits the closed-form solution

$$V_n(i,j)^* = \frac{[\text{sign}((\mu_j^n X_n(:,i)^\top Z(:,j))^2 - \gamma_j^n)]_+ \mu_j^n X_n(:,i)^\top Z(:,j)}{\sqrt{\sum_{k=1}^{\prod_{p \neq n} J_p} [\text{sign}((\mu_j^n X_n(:,k)^\top Z(:,j))^2 - \gamma_j^n)]_+ (\mu_j^n X_n(:,k)^\top Z(:,j))^2}}. \tag{27}$$

The final block reformulation is therefore

$$\psi_{\ell_0, R_n}^n(\gamma^n) = \max_{Z \in S_{R_n}^{J_n}} \sum_{j=1}^{R_n} \sum_{i=1}^{\prod_{k \neq n} J_k} [(\mu_j^n X_n(:,i)^\top Z(:,j))^2 - \gamma_j^n]_+, \tag{28}$$

which, like the $\ell_1$ case, maximizes a convex function on the Stiefel manifold $S_{R_n}^{J_n}$ and admits efficient optimization within the generalized power method framework.

# D ALGORITHM FOR GENERALIZED GRADIENT SCHEME

A reasonable stopping criterion for Algorithm 2 is to terminate either when the relative change of the objective function becomes sufficiently small,

$$\frac{f(x_{k+1}) - f(x_k)}{f(x_k)} \leq \epsilon, \tag{29}$$

or when the maximum number of iterations $k_{\max}$ is reached.

---

**Algorithm 2:** Generalized gradient scheme for convex maximization

---

**Input:** Initial iterate $x_0 \in Q$.
**Output:** Iterates $\{x_k\}$ approximating the solution of (14).

**1 while** *stopping criterion not met* **do**
**2** $\quad \big\lfloor \; x_{k+1} \leftarrow \arg\max_{y \in Q}\{f(x_k) + \langle f'(x_k), y - x_k\rangle\};$

---

# E  PROOF OF THEOREM 3.2

This appendix provides the detailed proof of Theorem 3.2.

*Proof.*  Let $\Delta(x) \overset{\text{def}}{=} \max_{y \in Q}\langle f'(x), y - x\rangle$. We first establish the estimate

$$\Delta(x_k) \geq \tfrac{\sigma_Q}{2}\|f'(x_k)\|_* \|x_{k+1} - x_k\|^2. \tag{30}$$

Since $f$ is convex, $\Delta(x_k) \geq 0$. Thus we focus on the case $\sigma_Q > 0$ and $f'(x_k) \neq 0$. By optimality of $x_{k+1}$ we have

$$\langle f'(x_k), y - x_{k+1}\rangle \geq 0 \quad \text{for all } y \in Conv(Q). \tag{31}$$

Choosing

$$y = y_\alpha \overset{\text{def}}{=} x_k + \alpha(x_{k+1} - x_k) + \tfrac{\sigma_Q}{2}\alpha(1-\alpha)\|x_{k+1} - x_k\|^2 \frac{G^{-1} f'(x_k)}{\|f'(x_k)\|_*}, \quad \alpha \in [0,1], \tag{32}$$

and using the definition of strong convexity of $Conv(Q)$ (Assumption 3), we obtain

$$0 \geq \langle f'(x_k), y_\alpha - x_{k+1}\rangle = (1-\alpha)\langle f'(x_k), x_k - x_{k+1}\rangle + \tfrac{\sigma_Q}{2}\alpha(1-\alpha)\|x_{k+1} - x_k\|^2 \|f'(x_k)\|_*. \tag{33}$$

Since $\alpha$ is arbitrary in $[0,1]$, the estimate follows. Here $G^{-1}$ denotes the inverse of the Riesz map associated with the chosen norm, which maps subgradients from the dual space back into $E$; in the Euclidean case, this reduces to identity, $G = I$, and thus $G^{-1} = I$.

Finally,

$$f(x_{k+1}) - f(x_k) \geq \Delta(x_k) + \tfrac{\sigma_f}{2}\|x_{k+1} - x_k\|^2 \geq \tfrac{\sigma_Q \delta_f + \sigma_f}{2}\|x_{k+1} - x_k\|^2, \tag{34}$$

and the additional assumptions ensure $\sigma_Q \delta_f + \sigma_f > 0$. Summing over $k \geq 0$ completes the proof.

$\square$

# F  ALGORITHMS FOR SPARSE PCA SUBPROBLEM WITH $\ell_1$ PENALTY

---

**Algorithm 3:** $\ell_1$ single-unit pattern finding

---

**Input:** $X_n \in \mathbb{R}^{J_n \times \prod_{i \neq n} J_i}$; parameter $\gamma^n \geq 0$; initial $z \in S_1^{J_n}$.
**Output:** Locally optimal sparsity pattern $P$.

**1 while** *stopping criterion not met* **do**
**2** $\quad \big\lceil \; z \leftarrow \sum_{i=1}^{\prod_{k \neq n} J_k}[\,|X_n(:,i)^\top z| - \gamma^n]_+ \operatorname{sign}(X_n(:,i)^\top z)\, X_n(:,i);$
**3** $\quad \big\lfloor \; z \leftarrow z/\|z\|_2;$
**4 Construct** $P \in \{0,1\}^{\prod_{i \neq n} J_i}$ with $P(i) = 1$ if $|X_n(:,i)^\top z| > \gamma^n$, else 0;

---

At each iteration, Algorithm 2 requires maximizing a convex function over the Stiefel manifold. As introduced in Section 2.1, we denote by $\operatorname{Polar}(A)$ the $U$ factor in the polar decomposition of a matrix $A \in \mathbb{R}^{p \times m}$,

$$A = UP, \tag{35}$$

where $U \in S_m^p$ and $P \in \mathbb{R}^{m \times m}$ is positive semi-definite. The polar decomposition has complexity $\mathcal{O}(pm^2)$ for $p \geq m$. By part of the Theorem 3.3, the main step of Algorithm 2 can be expressed as

$$x_{k+1} = \operatorname{Polar}(f'(x_k)). \tag{36}$$

This formulation directly yields Algorithm 3 and Algorithm 4.

For Algorithms 3 and 4, the stopping criterion is adopted from Appendix D.

---

**Algorithm 4:** $\ell_1$ block pattern finding

---

**Input:** $X_n \in \mathbb{R}^{J_n \times \prod_{i \neq n} J_i}$; parameters $\gamma_j^n \geq 0$, $\mu_j^n > 0$; initial $Z \in S_{R_n}^{J_n}$.
**Output:** Locally optimal sparsity pattern $P$.

**1 while** *stopping criterion not met* **do**

**2**     **for** $j = 1$ **to** $R_n$ **do**

**3**        $Z(:,j) \leftarrow \sum_{i=1}^{\prod_{k \neq n} J_k} \mu_j^n [\mu_j^n |X_n(:,i)^\top Z(:,j)| - \gamma_j^n]_+ \operatorname{sign}(X_n(:,i)^\top Z(:,j)) X_n(:,i)$;

**4**     $Z \leftarrow \operatorname{Polar}(Z)$;

**5 Construct** $P \in \{0,1\}^{\prod_{i \neq n} J_i \times R_n}$ with $P(i,j) = 1$ if $\mu_j^n |X_n(:,i)^\top Z(:,j)| > \gamma_j^n$, else 0;

---

## G  Algorithms for Sparse PCA Subproblem with $\ell_0$ penalty

The $\ell_0$-penalized counterparts of the algorithms in Section 3.7 are summarized below. These follow the same two-stage structure of pattern-finding and post-processing, but enforce sparsity directly through cardinality penalties. For parameter selection, the thresholds are chosen below the following natural upper bounds: $\gamma^n \leq \max_i \|X_n(:,i)\|_2^2$ for the single-unit case, and $\gamma_j^n \leq \max_i (\mu_j^n)^2 \|X_n(:,i)\|_2^2$ for the block case.

The computational complexity remains the same order as in the $\ell_1$ case: the single-unit algorithms (Algorithms 5) require $O\left(\prod_{n=1}^{N} J_n\right)$ operations per iteration, while the block algorithms (Algorithms 6) require $O\left(R_n \prod_{n=1}^{N} J_n\right)$ operations.

---

**Algorithm 5:** $\ell_0$ single-unit pattern finding

---

**Input:** $X_n \in \mathbb{R}^{J_n \times \prod_{i \neq n} J_i}$; parameter $\gamma^n \geq 0$; initial $z \in S_1^{J_n}$.
**Output:** Locally optimal sparsity pattern $P$.

**1 while** *stopping criterion not met* **do**

**2**     $z \leftarrow \sum_{i=1}^{\prod_{k \neq n} J_k} [\operatorname{sign}(X_n(:,i)^\top z)^2 - \gamma^n]_+ X_n(:,i)^\top z \, X_n(:,i)$;

**3**     $z \leftarrow z/\|z\|_2$;

**4 Construct** $P \in \{0,1\}^{\prod_{i \neq n} J_i}$ with $P(i) = 1$ if $(X_n(:,i)^\top z)^2 > \gamma^n$, else 0;

---

Following the discussion for the $\ell_1$-penalized case, the gradient-scheme update under $\ell_0$ penalties is also performed via the polar decomposition. In particular, the main iteration can be expressed as

$$x_{k+1} = \operatorname{Polar}\big(f'(x_k)\big), \tag{37}$$

This formulation naturally extends the $\ell_1$ setting and directly leads to Algorithm 5 and Algorithm 6.

---

**Algorithm 6:** $\ell_0$ block pattern finding

---

**Input:** $X_n \in \mathbb{R}^{J_n \times \prod_{i \neq n} J_i}$; parameters $\gamma_j^n \geq 0$, $\mu_j^n > 0$; initial $Z \in S_{R_n}^{J_n}$.
**Output:** Locally optimal sparsity pattern $P$.

**1 while** *stopping criterion not met* **do**

**2**     **for** $j = 1$ **to** $R_n$ **do**

**3**        $Z(:,j) \leftarrow \sum_{i=1}^{\prod_{k \neq n} J_k} (\mu_j^n)^2 [\operatorname{sign}(\mu_j^n X_n(:,i)^\top Z(:,j))^2 - \gamma_j^n]_+ X_n(:,i)^\top Z(:,j) \, X_n(:,i)$;

**4**     $Z \leftarrow \operatorname{Polar}(Z)$;

**5 Construct** $P \in \{0,1\}^{\prod_{i \neq n} J_i \times R_n}$ with $P(i,j) = 1$ if $(\mu_j^n X_n(:,i)^\top Z(:,j))^2 > \gamma_j^n$, else 0;

---

For Algorithms 5 and 6, the stopping criterion is adopted from Appendix D.

## H    PROOF OF THEOREM 3.3

This appendix provides the detailed proof of Theorem 3.3.

*Proof.*    **Step 1. Fixed $V_n$.** For $\mathrm{diag}(V_n{}^\top V_n) = I_{R_n}$, Problem (18) reduces to

$$\max_{Z \in S_{R_n}^{J_n}} \mathrm{Tr}(Z^\top X_n V_n N^n) = \max_{Z \in S_{R_n}^{J_n}} \langle Z, X_n V_n N^n \rangle. \tag{38}$$

Let $X_n V_n N^n = U\Sigma W^\top$ be the singular value decomposition (SVD), where $U$ is $J_n \times J_n$ orthonormal, $W$ is $R_n \times R_n$ orthonormal, and $\Sigma$ is diagonal with entries $\{\sigma_i\}_{i=1}^{R_n}$. Then

$$\max_{Z \in S_{R_n}^{J_n}} \langle Z, X_n V_n N^n \rangle = \max_{Z \in S_{R_n}^{J_n}} \langle U^\top Z W, \Sigma \rangle \leq \sum_{i=1}^{R_n} \sigma_i.$$

From the SVD, both factors of the polar decomposition are explicit. Setting $U'$ as the first $R_n$ columns of $U$ and $\Sigma'$ as the $R_n \times R_n$ principal block of $\Sigma$, we have

$$X_n V_n N^n = U'\Sigma'W^\top = (U'W^\top)(W\Sigma'W^\top). \tag{39}$$

Thus the polar decomposition gives $V = U'W^\top$ and $P = W\Sigma'W^\top$, with

$$\langle V, X_n V_n N^n \rangle = \mathrm{Tr}(P) = \sum_{i=1}^{R_n} \sigma_i. \tag{40}$$

Since $N^{n\top} V_n{}^\top X_n^\top X_n V_n N^n = P^2$, we obtain

$$Z^* = X_n V_n N^n \, (N^{n\top} V_n{}^\top X_n^\top X_n V_n N^n)^{-1/2}, \tag{41}$$

which is precisely the left orthonormal factor of the polar decomposition of $X_n V_n N^n$.

**Step 2. Fixed $Z$.** For $Z \in S_{R_n}^{J_n}$, Problem (18) becomes

$$\max_{\mathrm{diag}(V_n{}^\top V_n)=I_{R_n},\, V_n|_{P'}=0} \mathrm{Tr}(Z^\top X_n V_n N^n). \tag{42}$$

The Lagrangian of the optimization Problem (42) is

$$\mathcal{L}(V_n, \Lambda_1, \Lambda_2) = \mathrm{Tr}(Z^\top X_n V_n N^n) - \mathrm{Tr}(\Lambda_1(V_n{}^\top V_n - I_{R_n})) - \mathrm{Tr}(\Lambda_2^\top V_n), \tag{43}$$

where $\Lambda_1$ is diagonal and invertible, and $\Lambda_2|_P = 0$. The first-order conditions yield

$$X_n^\top Z N^n - 2V_n \Lambda_1 - \Lambda_2 = 0, \qquad \mathrm{diag}(V_n{}^\top V_n) = I_{R_n}, \qquad V_n|_{P'} = 0. \tag{44}$$

Hence any stationary point $V_n$ satisfies

$$V_n|_P = (X_n^\top Z N^n D)|_P, \qquad V_n|_{P'} = 0, \tag{45}$$

where $D$ is a positive diagonal matrix normalizing the columns of $V_n$, explicitly

$$D = \mathrm{diag}(N^n Z^\top X_n X_n^\top Z N^n)^{-1/2}. \tag{46}$$

$\square$

## I    ALGORITHM FOR POST-PROCESSING

For Algorithms 7, the stopping criterion is adopted from Appendix D.

---

**Algorithm 7:** Alternating optimization scheme for post-processing

---

**Input:** Data matrix $X_n \in \mathbb{R}^{J_n \times \prod_{i \neq n} J_i}$; sparsity pattern $P$; diagonal weight matrix
$N^n = \mathrm{diag}(\mu_1, \ldots, \mu_{R_n})$; initial iterate $Z \in S_{R_n}^{J_n}$.
**Output:** Local maximizer $(Z, V_n)$ of (16).

**1 while** *stopping criterion not met* **do**
**2**     $V_n \leftarrow X_n^\top Z N^n$;
**3**     $V_n \leftarrow V_n \, \mathrm{diag}(V_n^\top V_n)^{-1/2}$;
**4**     $V_n|_{P'} \leftarrow 0$;
**5**     $Z \leftarrow \mathrm{Polar}(X_n V_n N^n)$;

---

Table 2: Construction of $V_n|_P$.

|  | Computation of $P$ | Computation of $V_n|_P$ |
|---|---|---|
| GP-STPCA($\ell_1$) | Algorithm 3 | Solution (17) |
| GP-STPCA($\ell_0$) | Algorithm 5 | Solution (23) |
| GP-STPCA($\ell_1$, block) | Algorithm 4 | Algorithm 7 |
| GP-STPCA($\ell_0$, block) | Algorithm 6 | Solution (27) |

Table 3: Summary of the four GP-STPCA variants and their per-iteration computational complexity.

|  | **Algorithm** | **Complexity (per iteration)** |
|---|---|---|
| GP-STPCA($\ell_1$) | Algorithm 8 | $\mathcal{O}\left(\prod_{n=1}^N J_n\right)$ |
| GP-STPCA($\ell_1$, block) | Algorithm 9 | $\mathcal{O}\left(R_n \prod_{n=1}^N J_n\right)$ |
| GP-STPCA($\ell_0$) | Algorithm 10 | $\mathcal{O}\left(\prod_{n=1}^N J_n\right)$ |
| GP-STPCA($\ell_0$, block) | Algorithm 11 | $\mathcal{O}\left(R_n \prod_{n=1}^N J_n\right)$ |

---

**Algorithm 8:** GP-STPCA($\ell_1$)

---

**Input:** Tensor $\mathcal{X} \in \mathbb{R}^{J_1 \times J_2 \times \cdots \times J_N}$; target ranks $(R_1, \ldots, R_N)$.
**Output:** Factor matrices $\{U_n\}_{n=1}^N$ and core tensor $\mathcal{G}$.

**1 for** $n = 1$ *to* $N$ **do**
**2**     Compute $X_n = X_{(n)} \in \mathbb{R}^{J_n \times \prod_{i \neq n} J_i}$;
**3**     **for** $r = 1$ *to* $R_n$ **do**
**4**        Initialize the parameter $\gamma^n \geq 0$ and the vector $z \in S_1^{J_n}$;
**5**        **while** *stopping criterion not met* **do**
**6**           $z \leftarrow \sum_{i=1}^{\prod_{k \neq n} J_k} [\, |X_n(:,i)^\top z| - \gamma^n \,]_+ \, \mathrm{sign}(X_n(:,i)^\top z) \, X_n(:,i)$;
**7**           $z \leftarrow z/\|z\|_2$;
**8**        Construct $P \in \{0,1\}^{\prod_{i \neq n} J_i}$ with $P(i) = 1$ if $|X_n(:,i)^\top z| > \gamma^n$, else 0;
**9**        Compute rank-1 decomposition $X_n|_P = \sigma u v^\top$;
**10**       $v_r \leftarrow v$;
**11**       $z \leftarrow \min_z \|X_n - z v_r^\top\|_F$;
**12**       $X_n \leftarrow X_n - z v_r^\top$;
**13**     Construct $V_n$ by all $v_r$;
**14**     Compute $U_n$ as the left orthonormal factor in the SVD of $X_n V_n$;
**15**     Set $U_n \in \mathbb{R}^{J_n \times R_n}$ as the final solution for mode $n$;
**16** Compute the core tensor as: $\mathcal{G} = \mathcal{X} \times_1 U_1^\top \times_2 U_2^\top \cdots \times_N U_N^\top$.

---

---

**Algorithm 9:** GP-STPCA($\ell_1$, block)

**Input:** Tensor $\mathcal{X} \in \mathbb{R}^{J_1 \times J_2 \times \cdots \times J_N}$; target ranks $(R_1, \ldots, R_N)$.
**Output:** Factor matrices $\{U_n\}_{n=1}^{N}$ and core tensor $\mathcal{G}$.

**1 for** $n = 1$ *to* $N$ **do**

   **2**    Compute $X_n = X_{(n)} \in \mathbb{R}^{J_n \times \Pi_{i \neq n} J_i}$;

   **3**    Initialize the parameters $\gamma_j^n \geq 0$, $\mu_j^n > 0$ and the matrix $Z \in S_{R_n}^{J_n}$;

   **4**    **while** *stopping criterion not met* **do**

     **5**     **for** $j = 1$ **to** $R_n$ **do**

      **6**      $Z(:,j) \leftarrow \sum_{i=1}^{\Pi_{k \neq n} J_k} \mu_j^n [\mu_j^n |X_n(:,i)^\top Z(:,j)| - \gamma_j^n]_+ \operatorname{sign}(X_n(:,i)^\top Z(:,j)) X_n(:,i)$;

   **7**    $Z \leftarrow \operatorname{Polar}(Z)$;

   **8**    Construct $P \in \{0,1\}^{\Pi_{i \neq n} J_i \times R_n}$ with $P(i,j) = 1$ if $\mu_j^n |X_n(:,i)^\top Z(:,j)| > \gamma_j^n$, else 0;

   **9**    **while** *stopping criterion not met* **do**

   **10**     $V_n \leftarrow X_n^\top Z N^n$;

   **11**     $V_n \leftarrow V_n \operatorname{diag}(V_n^\top V_n)^{-1/2}$;

   **12**     $V_n|_{P'} \leftarrow 0$ ($P'$ is the complement of $P$);

   **13**     $Z \leftarrow \operatorname{Polar}(X_n V_n N^n)$;

   **14**    Compute $U_n$ as the left orthonormal factor in the SVD of $X_n V_n$;

   **15**    Set $U_n \in \mathbb{R}^{J_n \times R_n}$ as the final solution for mode $n$;

**16** Compute the core tensor as: $\mathcal{G} = \mathcal{X} \times_1 U_1^\top \times_2 U_2^\top \cdots \times_N U_N^\top$.

---

**Algorithm 10:** GP-STPCA($\ell_0$)

**Input:** Tensor $\mathcal{X} \in \mathbb{R}^{J_1 \times J_2 \times \cdots \times J_N}$; target ranks $(R_1, \ldots, R_N)$.
**Output:** Factor matrices $\{U_n\}_{n=1}^{N}$ and core tensor $\mathcal{G}$.

**1 for** $n = 1$ *to* $N$ **do**

   **2**    Compute $X_n = X_{(n)} \in \mathbb{R}^{J_n \times \Pi_{i \neq n} J_i}$;

   **3**    **for** $r = 1$ *to* $R_n$ **do**

     **4**     Initialize the parameter $\gamma^n \geq 0$ and the vector $z \in S_1^{J_n}$;

     **5**     **while** *stopping criterion not met* **do**

      **6**      $z \leftarrow \sum_{i=1}^{\Pi_{k \neq n} J_k} [\operatorname{sign}(X_n(:,i)^\top z)^2 - \gamma^n]_+ X_n(:,i)^\top z \, X_n(:,i)$; $z \leftarrow z/\|z\|_2$;

     **7**     Construct $P \in \{0,1\}^{\Pi_{i \neq n} J_i}$ with $P(i) = 1$ if $(X_n(:,i)^\top z)^2 > \gamma^n$, else 0;

     **8**     **for** $i = 1$ *to* $\prod_{k \neq n} J_k$ **do**

      **9**      **while** $P(i) = 1$ **do**

      **10**       $v_r(i) = \dfrac{[\operatorname{sign}((X_n(:,i)^\top z)^2 - \gamma^n)]_+ X_n(:,i)^\top z}{\sqrt{\sum_{k=1}^{\Pi_{p \neq n} J_p} [\operatorname{sign}((X_n(:,i)^\top z)^2 - \gamma^n)]_+ (X_n(:,i)^\top z)^2}}$.

     **11**     $z \leftarrow \min_z \|X_n - z v_r^\top\|_F$;

     **12**     $X_n \leftarrow X_n - z v_r^\top$;

   **13**    Construct $V_n$ by all $v_r$;

   **14**    Compute $U_n$ as the left orthonormal factor in the SVD of $X_n V_n$;

   **15**    Set $U_n \in \mathbb{R}^{J_n \times R_n}$ as the final solution for mode $n$;

**16** Compute the core tensor as: $\mathcal{G} = \mathcal{X} \times_1 U_1^\top \times_2 U_2^\top \cdots \times_N U_N^\top$.

---

---

**Algorithm 11:** GP-STPCA($\ell_0$, block)

---

**Input:** Tensor $\mathcal{X} \in \mathbb{R}^{J_1 \times J_2 \times \cdots \times J_N}$; target ranks $(R_1, \ldots, R_N)$.
**Output:** Factor matrices $\{U_n\}_{n=1}^N$ and core tensor $\mathcal{G}$.

**1 for** $n = 1$ *to* $N$ **do**

**2**     Compute $X_n = X_{(n)} \in \mathbb{R}^{J_n \times \prod_{i \neq n} J_i}$;

**3**     Initialize the parameters $\gamma_j^n \geq 0$, $\mu_j^n > 0$ and the matrix $Z \in S_{R_n}^{J_n}$;

**4**     **while** *stopping criterion not met* **do**

**5**        **for** $j = 1$ **to** $R_n$ **do**

**6**           $Z(:,j) \leftarrow \sum_{i=1}^{\prod_{k \neq n} J_k} (\mu_j^n)^2 [\operatorname{sign}(\mu_j^n X_n(:,i)^\top Z(:,j))^2 - \gamma_j^n]_+ X_n(:,i)^\top Z(:,j) X_n(:,i)$;

**7**        $Z \leftarrow \operatorname{Polar}(Z)$;

**8**     Construct $P \in \{0,1\}^{\prod_{i \neq n} J_i \times R_n}$ with $P(i,j) = 1$ if $(\mu_j^n X_n(:,i)^\top Z(:,j))^2 > \gamma_j^n$, else 0;

**9**     **for** $i = 1$ **to** $\prod_{k \neq n} J_k$ **do**

**10**        **for** $j = 1$ **to** $R_n$ **do**

**11**           **while** $P(i,j) = 1$ **do**

**12**              $V_n(i,j) = \dfrac{[\operatorname{sign}((\mu_j^n X_n(:,i)^\top Z(:,j))^2 - \gamma_j^n)]_+ \mu_j^n X_n(:,i)^\top Z(:,j)}{\sqrt{\sum_{k=1}^{\prod_{p \neq n} J_p} [\operatorname{sign}((\mu_j^n X_n(:,k)^\top Z(:,j))^2 - \gamma_j^n)]_+ (\mu_j^n X_n(:,k)^\top Z(:,j))^2}}$;

**13**     Compute $U_n$ as the left orthonormal factor in the SVD of $X_n V_n$;

**14**     Set $U_n \in \mathbb{R}^{J_n \times R_n}$ as the final solution for mode $n$;

**15** Compute the core tensor as: $\mathcal{G} = \mathcal{X} \times_1 U_1^\top \times_2 U_2^\top \cdots \times_N U_N^\top$.

---

## J    DETAILS OF GP-STPCA AND INITIALIZATION OF HYPERPARAMETERS

To summarize, we present four variants for constructing $V_n$, which combine a procedure for identifying a suitable sparsity pattern $P$ with a corresponding method for computing the active entries. The specific combinations are listed in Table 2.

We are now ready to present the complete algorithms for the four variants of GP-STPCA. Algorithm 8 corresponds to the $\ell_1$ single-unit formulation, Algorithm 9 to the $\ell_1$ block formulation, Algorithm 10 to the $\ell_0$ single-unit formulation, and Algorithm 11 to the $\ell_0$ block formulation. Table 3 provides an overview of the four variants, where the last column lists the per-iteration computational complexity of the corresponding subproblems, derived from the analyses in Section 3.7 and Appendix G. For all algorithms, the stopping criterion follows Appendix D.

Initialization must ensure at least one active element in the sparsity pattern. For the single-unit case, we choose $z \in S_1^{J_n}$ parallel to the column of $X_n$ with maximum norm:

$$z = \frac{X_n(:,i^*)}{\|X_n(:,i^*)\|_2}, \quad i^* = \arg\max_i \|X_n(:,i)\|_2. \tag{47}$$

For the block case, we set $Z = [z \,|\, Z_\perp]$ with $Z_\perp \in S_{R_n-1}^{J_n}$ orthogonal to $z$.

According to the analysis in the literature, the sparsity-inducing parameters can be initialized randomly within the following ranges:

- for GP-STPCA($\ell_1$), $\gamma^n \sim \mathcal{N}(0, \max_i \|X_n(:,i)\|_2)$;

- for GP-STPCA($\ell_0$), $\sqrt{\gamma^n} \sim \mathcal{N}(0, \max_i \|X_n(:,i)\|_2)$;

- for GP-STPCA($\ell_1$, block), $\gamma_j^n \sim \mathcal{N}\big(0, \mu_j^n \max_i \|X_n(:,i)\|_2\big)$.

- for GP-STPCA($\ell_0$, block), $\sqrt{\gamma_j^n} \sim \mathcal{N}\big(0, \mu_j^n \max_i \|X_n(:,i)\|_2\big)$;

Thus, in all four cases, the parameters may be drawn randomly from these distributions. However, as demonstrated in the experiments of Journée et al. (2010), the algorithms are generally robust to

Table 4: Choices of the hyperparameters $\gamma^n$ and $\gamma_j^n$ for different GP-STPCA variants.

| Algorithm | Hyperparameter |
|---|---|
| GP-STPCA($\ell_1$) | $\gamma^n = \frac{1}{2} \max_i \|X_n(:,i)\|_2$ |
| GP-STPCA($\ell_0$) | $\gamma^n = \frac{1}{4} \max_i \|X_n(:,i)\|_2^2$ |
| GP-STPCA($\ell_1$, block) | $\gamma_j^n = \frac{1}{2} \mu_j^n \max_i \|X_n(:,i)\|_2$ |
| GP-STPCA($\ell_0$, block) | $\gamma_j^n = \frac{1}{4} (\mu_j^n)^2 \max_i \|X_n(:,i)\|_2^2$ |

initialization. Hence, the fixed values listed in Table 4 serve as a reliable guideline for choosing initial parameters in practice.

For the block case, $N^n$ can be chosen as needed; in our experiments, we simply set it to the identity matrix.

## K    ADDITIONAL DETAILS ON SYNTHETIC EXPERIMENTS ON SPARSE SUPPORT RECOVERY

### K.1    SIMULATION SETUP

We design four simulation scenarios to capture both balanced and unbalanced settings with different sparsity structures:

- **Scenario 1 (balanced)**: $100 \times 100 \times 100$, sparsity only in mode $\mathbf{U}$;
- **Scenario 2 (unbalanced)**: $1000 \times 20 \times 20$, sparsity only in mode $\mathbf{U}$;
- **Scenario 3 (balanced)**: $100 \times 100 \times 100$, sparsity in all three modes;
- **Scenario 4 (unbalanced)**: $1000 \times 20 \times 20$, sparsity in all three modes.

For sparse modes, 50% of the entries are randomly set to zero and the remaining entries are sampled from $N(0,1)$. For dense modes, the factors are constructed as the leading $K$ left or right singular vectors of random Gaussian matrices with i.i.d. $N(0,1)$ entries. Scenarios 1 and 3 thus represent balanced tensors, while Scenarios 2 and 4 illustrate unbalanced cases with one dominant mode.

### K.2    ROC ANALYSIS AND FEATURE SELECTION ACCURACY

To quantify the accuracy of support recovery, we report averaged Receiver Operating Characteristic (ROC) curves over 50 replications in each simulation setting. Figure 2 displays ROC curves for mode-$u_1$ in Scenarios 1 and 2, where it is the only sparse mode. Figure 5 presents ROC curves for modes $u_1$, $v_1$, and $w_1$ in Scenarios 3 and 4, where sparsity is present in all modes.

In both settings, all variants of the proposed *GP-STPCA* achieve recovery performance comparable to *sparseGeoHOPCA* while significantly outperforming HOPCA (Kolda & Bader, 2009), and they do so with markedly faster computation.

As illustrated in Figures 2 and 5, *sparseGeoHOPCA* consistently maintains a high true positive rate while keeping the false positive rate relatively low across all simulation scenarios. Among the *GP-STPCA* variants, the choice of $\ell_0$ or $\ell_1$ penalty leads to similar results; however, the block formulations consistently yield larger areas under the ROC curve (AUC), highlighting their robustness and reliability. Overall, these results validate the effectiveness of *GP-STPCA* for sparse recovery in both balanced and unbalanced settings.

### K.3    SUPPORT RECOVERY AND EFFICIENCY COMPARISON

Table 5 reports the mean and standard deviation of true positive (TP) and false positive (FP) rates, together with runtime, for HOPCA, sparseGeoHOPCA, and the four GP-STPCA variants across Scenarios 1–4. The results complement the ROC analysis by providing a detailed quantitative comparison.

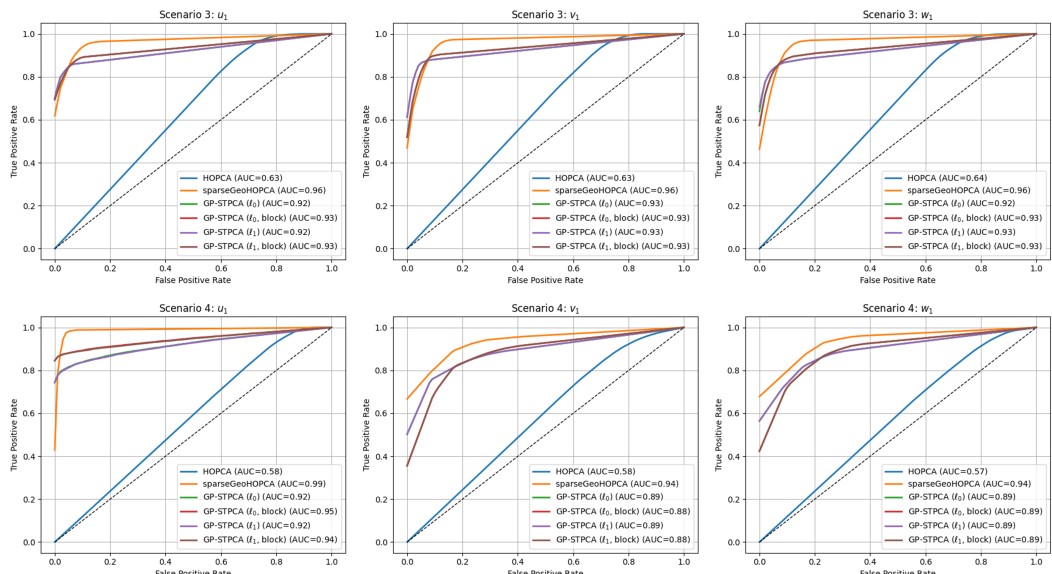

Figure 5: ROC curves for modes $u_1$, $v_1$, and $w_1$ in Scenarios 3 and 4, where sparsity is present in all modes. Results are averaged over fifty independent runs.

Overall, HOPCA attains perfect TP rates across all modes, but at the expense of excessive false positives, often above 70% in Scenarios 3 and 4, which indicates very poor feature specificity. SparseGeoHOPCA substantially alleviates this issue by maintaining high TP recovery while reducing FP to below 10% in most cases, albeit with high computational cost ($\sim$1–1.6s per run). By contrast, the proposed GP-STPCA variants achieve recovery performance competitive with sparseGeo-HOPCA but with drastically reduced runtime (typically $\sim$0.01–0.02s). Differences between $\ell_0$- and $\ell_1$-penalties are minor, while block variants generally yield slightly higher TP rates at the cost of somewhat larger FP in certain modes (e.g., Scenario 4, mode $u_1$).

In summary, HOPCA provides high sensitivity but fails to control false positives; sparseGeoHOPCA achieves accurate and stable recovery but is computationally expensive; GP-STPCA, in contrast, combines the advantages of both—offering reliable support recovery comparable to sparseGeo-HOPCA while being orders of magnitude faster. This balance of accuracy, sparsity control, and efficiency makes GP-STPCA a practical and scalable choice for large-scale sparse tensor decomposition tasks.

## L   DETAILS OF IMAGE RECONSTRUCTION EXPERIMENT

**Experimental protocol.**   We evaluate the proposed *GP-STPCA* on six ImageNet RGB images using four variants (penalty and block settings): GP-STPCA ($\ell_0$), GP-STPCA ($\ell_0$, block), GP-STPCA ($\ell_1$), and GP-STPCA ($\ell_1$, block). We randomly selected six RGB images from the ImageNet dataset Russakovsky et al. (2015), with original resolutions of $446 \times 349 \times 3$, $472 \times 349 \times 3$, $472 \times 349 \times 3$, $472 \times 349 \times 3$, $472 \times 349 \times 3$, and $349 \times 349 \times 3$, respectively. The baseline *sparseGeoHOPCA* is included for comparison. All methods retain the same number of components (90), and the reconstructions are obtained by linearly combining the selected bases. Representative visuals are shown in Figure 3 (Image1–3) and Figure 6 (Image4–6); the corresponding PSNR tables are Table 1 and Table 6, respectively.

**Qualitative results (Figs. 3 and 6).**   Across all six samples, GP-STPCA variants produce sharper edges and more faithful textures than the baseline. Typical degradations observed in the baseline—blurred details, color wash-out, and faint directional banding—are substantially reduced by GP-STPCA. Block variants are particularly effective on scenes with complex local textures (e.g., the bird feathers, coral/fish patterns, and sailboat rigging in Figure 3), where they better preserve fine structures and suppress streaking.

Table 5: Support recovery (TP/FP) and runtime for Scenarios 1–4.

| Scenario | Mode | Method | TP (mean ± std) | FP (mean ± std) | Time (s) |
|---|---|---|---|---|---|
| 1 | $u_1$ | HOPCA | 1.000 ± 0.000 | 0.486 ± 0.116 | 0.136 |
| | | sparseGeoHOPCA | 0.967 ± 0.046 | 0.029 ± 0.046 | 1.038 |
| | | GP-STPCA ($\ell_0$) | 0.857 ± 0.065 | 0.007 ± 0.020 | 0.018 |
| | | GP-STPCA ($\ell_0$, block) | 0.888 ± 0.060 | 0.015 ± 0.031 | 0.018 |
| | | GP-STPCA ($\ell_1$) | 0.857 ± 0.065 | 0.007 ± 0.020 | 0.021 |
| | | GP-STPCA ($\ell_1$, block) | 0.888 ± 0.060 | 0.015 ± 0.031 | 0.021 |
| 2 | $u_1$ | HOPCA | 1.000 ± 0.000 | 0.673 ± 0.064 | 0.076 |
| | | sparseGeoHOPCA | 0.989 ± 0.014 | 0.008 ± 0.011 | 1.622 |
| | | GP-STPCA ($\ell_0$) | 0.976 ± 0.061 | 0.561 ± 0.021 | 0.009 |
| | | GP-STPCA ($\ell_0$, block) | 0.989 ± 0.030 | 0.644 ± 0.020 | 0.008 |
| | | GP-STPCA ($\ell_1$) | 0.975 ± 0.062 | 0.569 ± 0.022 | 0.010 |
| | | GP-STPCA ($\ell_1$, block) | 0.988 ± 0.033 | 0.642 ± 0.021 | 0.010 |
| 3 | $u_1$ | HOPCA | 1.000 ± 0.000 | 0.734 ± 0.079 | 0.141 |
| | | sparseGeoHOPCA | 0.958 ± 0.053 | 0.028 ± 0.045 | 1.041 |
| | | GP-STPCA ($\ell_0$) | 0.850 ± 0.075 | 0.006 ± 0.017 | 0.020 |
| | | GP-STPCA ($\ell_0$, block) | 0.881 ± 0.070 | 0.013 ± 0.029 | 0.019 |
| | | GP-STPCA ($\ell_1$) | 0.850 ± 0.075 | 0.006 ± 0.017 | 0.020 |
| | | GP-STPCA ($\ell_1$, block) | 0.881 ± 0.070 | 0.013 ± 0.029 | 0.020 |
| | $v_1$ | HOPCA | 1.000 ± 0.000 | 0.736 ± 0.077 | 0.141 |
| | | sparseGeoHOPCA | 0.968 ± 0.048 | 0.043 ± 0.052 | 1.041 |
| | | GP-STPCA ($\ell_0$) | 0.869 ± 0.077 | 0.011 ± 0.020 | 0.020 |
| | | GP-STPCA ($\ell_0$, block) | 0.893 ± 0.069 | 0.024 ± 0.034 | 0.019 |
| | | GP-STPCA ($\ell_1$) | 0.869 ± 0.077 | 0.011 ± 0.020 | 0.020 |
| | | GP-STPCA ($\ell_1$, block) | 0.893 ± 0.069 | 0.024 ± 0.034 | 0.020 |
| | $w_1$ | HOPCA | 1.000 ± 0.000 | 0.730 ± 0.077 | 0.141 |
| | | sparseGeoHOPCA | 0.963 ± 0.058 | 0.045 ± 0.053 | 1.041 |
| | | GP-STPCA ($\ell_0$) | 0.862 ± 0.085 | 0.013 ± 0.031 | 0.020 |
| | | GP-STPCA ($\ell_0$, block) | 0.889 ± 0.080 | 0.024 ± 0.042 | 0.019 |
| | | GP-STPCA ($\ell_1$) | 0.863 ± 0.085 | 0.012 ± 0.031 | 0.020 |
| | | GP-STPCA ($\ell_1$, block) | 0.889 ± 0.080 | 0.024 ± 0.042 | 0.020 |
| 4 | $u_1$ | HOPCA | 1.000 ± 0.000 | 0.850 ± 0.071 | 0.073 |
| | | sparseGeoHOPCA | 0.987 ± 0.016 | 0.012 ± 0.017 | 1.644 |
| | | GP-STPCA ($\ell_0$) | 0.942 ± 0.110 | 0.576 ± 0.018 | 0.009 |
| | | GP-STPCA ($\ell_0$, block) | 0.977 ± 0.052 | 0.753 ± 0.019 | 0.008 |
| | | GP-STPCA ($\ell_1$) | 0.942 ± 0.110 | 0.579 ± 0.023 | 0.009 |
| | | GP-STPCA ($\ell_1$, block) | 0.977 ± 0.057 | 0.750 ± 0.020 | 0.009 |
| | $v_1$ | HOPCA | 1.000 ± 0.000 | 0.841 ± 0.124 | 0.072 |
| | | sparseGeoHOPCA | 0.927 ± 0.096 | 0.056 ± 0.104 | 1.570 |
| | | GP-STPCA ($\ell_0$) | 0.844 ± 0.123 | 0.066 ± 0.106 | 0.011 |
| | | GP-STPCA ($\ell_0$, block) | 0.880 ± 0.108 | 0.117 ± 0.130 | 0.011 |
| | | GP-STPCA ($\ell_1$) | 0.844 ± 0.123 | 0.066 ± 0.106 | 0.012 |
| | | GP-STPCA ($\ell_1$, block) | 0.880 ± 0.108 | 0.117 ± 0.130 | 0.012 |
| | $w_1$ | HOPCA | 1.000 ± 0.000 | 0.859 ± 0.109 | 0.072 |
| | | sparseGeoHOPCA | 0.938 ± 0.085 | 0.064 ± 0.113 | 1.570 |
| | | GP-STPCA ($\ell_0$) | 0.856 ± 0.119 | 0.068 ± 0.110 | 0.011 |
| | | GP-STPCA ($\ell_0$, block) | 0.898 ± 0.108 | 0.112 ± 0.135 | 0.011 |
| | | GP-STPCA ($\ell_1$) | 0.856 ± 0.119 | 0.068 ± 0.110 | 0.012 |
| | | GP-STPCA ($\ell_1$, block) | 0.898 ± 0.108 | 0.112 ± 0.135 | 0.012 |

**Quantitative results (Tables 1 and 6).** On **Image1–3**, the best PSNR in each image is attained by a block variant: $\ell_0$, *block* on Image1 (27.692 dB) and $\ell_1$, *block* on Image2 (23.054 dB) and Image3 (31.727 dB). Relative to the baseline, the PSNR gains are substantial: $+6.20$ dB (Image1), $+2.09$ dB (Image2), and $+4.01$ dB (Image3). Averaged over Image1–3, GP-STPCA ($\ell_1$, block) achieves the highest mean PSNR (27.01 dB), exceeding the baseline by $+3.62$ dB.

On **Image4–6**, the best PSNRs are achieved by the non-block variants: $\ell_1$ on Image4 (29.055 dB) and $\ell_0$ on Image5 (17.735 dB) and Image6 (20.466 dB). Improvements over the baseline remain consistent and often large: $+5.55$ dB (Image4), $+0.74$ dB (Image5), and $+4.69$ dB (Image6). Averaged over Image4–6, GP-STPCA ($\ell_1$) attains the highest mean PSNR (21.61 dB), outperforming the baseline by about $+2.85$ dB.

**Conclusions.** (i) GP-STPCA dominates the baseline on every image, with per-image gains up to $+6.20$ dB and an overall mean improvement of $\approx +3.2$ dB across all six images. (ii) Block modeling matters for highly textured scenes (Image1–3), where local spatial correlations are strong; here, block variants—especially GP-STPCA ($\ell_1$, block)—consistently yield the top PSNR and visibly cleaner details. (iii) Penalty choice is data dependent: $\ell_0$ excels on scenes where retaining a few high-energy structures drives quality (Image5–6), while $\ell_1$ provides robust, high-PSNR reconstructions under more heterogeneous textures (Image4) and, in its block form, achieves the highest average on Image1–3. In practice, we recommend starting with GP-STPCA ($\ell_1$, block) for complex textures and switching to $\ell_0$ when sharp, sparse structures dominate.

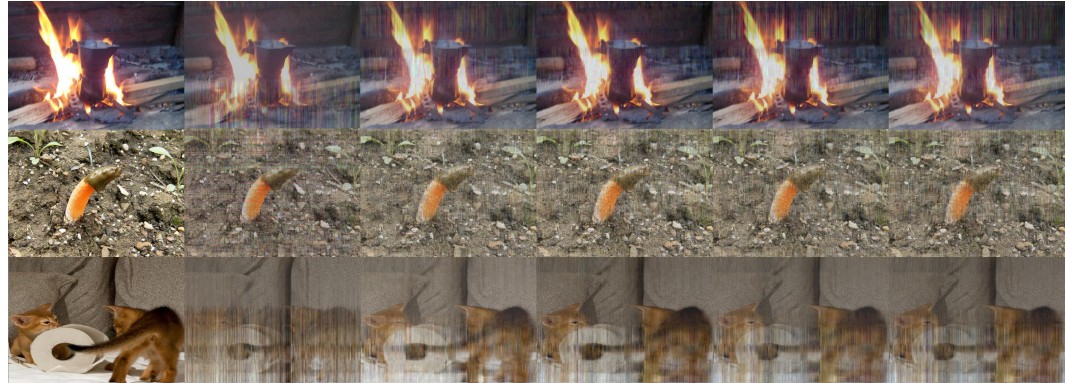

Figure 6: Visual comparison of ImageNet reconstructions: original, sparseGeoHOPCA, GT-STPCA($\ell_0$), GT-STPCA($\ell_0$, block), GT-STPCA($\ell_1$), and GT-STPCA($\ell_1$, block).

Table 6: PSNR (dB) comparison on Image4–6 in Figure 6.

| Method | Image4 | Image5 | Image6 |
|---|---|---|---|
| sparseGeoHOPCA | 23.507 | 16.999 | 15.773 |
| GP-STPCA ($\ell_0$) | 26.285 | **17.735** | **20.466** |
| GP-STPCA ($\ell_0$, block) | 25.338 | 17.503 | 19.351 |
| GP-STPCA ($\ell_1$) | **29.055** | 17.563 | 18.219 |
| GP-STPCA ($\ell_1$, block) | 25.105 | 17.410 | 17.746 |

## M  DETAILS OF CONNECTOME-BASED ANALYSIS OF BRAIN NETWORK

We investigate the relationship between brain structural connectomes and cognitive traits using data from the Human Connectome Project (HCP) (Van Essen et al., 2013). The full HCP dataset includes 1065 subjects with diffusion MRI scans (Zhang et al., 2019), from which structural connectivity networks were extracted. Each subject's connectome is represented as a $68 \times 68$ connected surface area (CSA) (Zhang et al., 2018) network, where nodes correspond to cortical regions defined by the popular Desikan–Killiany atlas (Desikan et al., 2006), and edge weights capture white-matter connectivity strength.

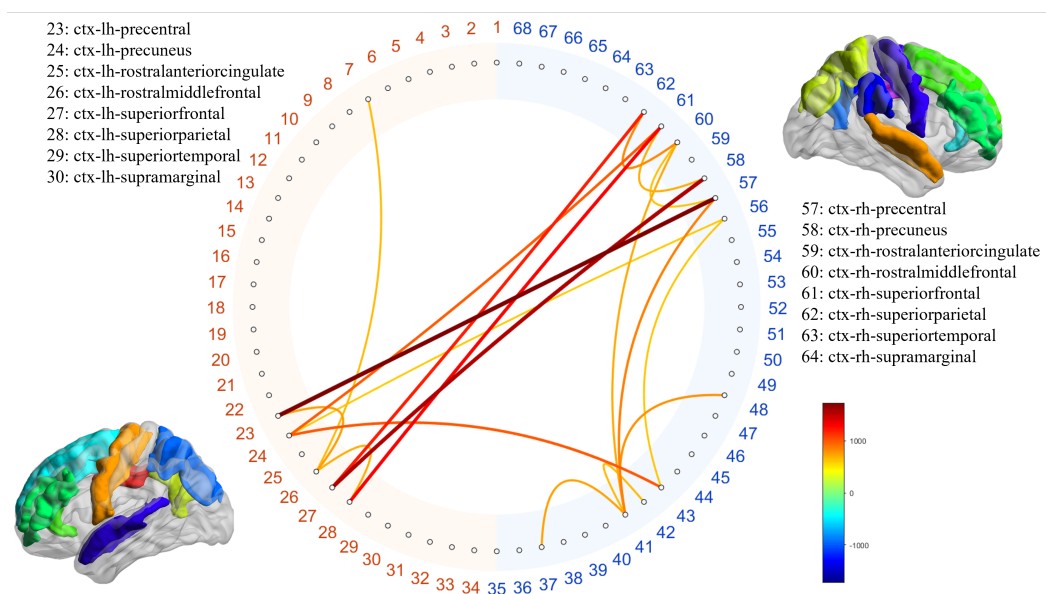

Figure 7: Chord diagram of enhanced connectivity in high-reading group.

To construct these networks, the cortex is parcellated into $68$ anatomical regions of interest (ROIs), $34$ per hemisphere. For each pair of ROIs, streamlines traced from diffusion tractography are used to determine the white-matter pathways linking them. To ensure meaningful connectivity measures, each gray matter ROI is slightly dilated to include a portion of the adjacent white matter, streamlines passing through multiple ROIs are segmented into appropriate portions, and apparent outliers are removed. The resulting $68 \times 68$ weighted adjacency matrix quantifies inter-regional coupling and provides a reproducible representation of each subject's structural connectome. Further details can be found in Zhang et al. (2018).

For each cognitive trait of interest, subjects are ranked according to their scores. Following the procedure in Zhang et al. (2019), we extract two groups of subjects: the $100$ individuals with the highest scores and the $100$ with the lowest scores. This yields a subset of $200$ subjects, each represented by a $68 \times 68$ CSA connectivity matrix, forming a three-order tensor $\mathcal{X}$ of size $68 \times 68 \times 200$. This tensor representation serves as the input for exploratory analyses, allowing us to identify discriminative connectivity patterns associated with the trait. In addition to imaging data, the HCP provides a broad set of behavioral and cognitive trait measures. Here, we focus on the age-adjusted English reading score as a representative cognitive trait.

We apply the proposed STPCA ($\ell_1$, block) to obtain low-dimensional principal component (PC) scores from the CSA networks. These PC scores, together with demographic covariates, are used for downstream statistical analysis. For continuous traits such as reading score, we employ canonical correlation analysis (CCA) to estimate a discriminant direction $U_3$ in the embedding space. To interpret the results in the original connectivity domain, we project $U_3$ back, and visualize the top-$50$ edges with the largest group differences to form a difference network $\Delta$net using a chord diagram, highlighting the subset of edges most correlated with the trait.

Figure 4 illustrates the top-50 edges in $\Delta$net with the largest differences between high- and low-reading groups. Warm colors (red–yellow) denote stronger connections in the high-reading group, while cool colors (blue) denote stronger connections in the low-reading group. Edge thickness reflects the magnitude of the difference. In the chord diagrams (Figures 7 and 8), we observe that subjects with low reading scores exhibit reduced long-range integration across fronto-parietal and fronto-temporal pathways. In contrast, the high-reading group shows enhanced connectivity in key language-related regions, underscoring the role of distributed cortical networks in supporting reading ability. Subjects with high reading scores exhibit enhanced inter-hemispheric connectivity across fronto-parietal circuits, while low-reading subjects show weaker integration. Importantly, no edges were found to decrease significantly with increasing reading ability.

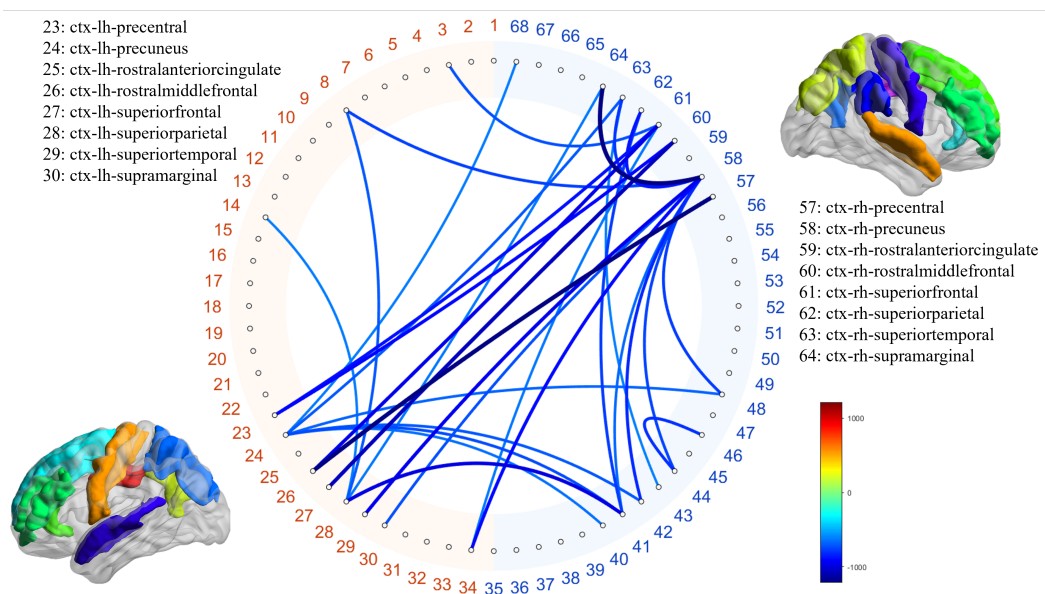

23: ctx-lh-precentral
24: ctx-lh-precuneus
25: ctx-lh-rostralanteriorcingulate
26: ctx-lh-rostralmiddlefrontal
27: ctx-lh-superiorfrontal
28: ctx-lh-superiorparietal
29: ctx-lh-superiortemporal
30: ctx-lh-supramarginal

57: ctx-rh-precentral
58: ctx-rh-precuneus
59: ctx-rh-rostralanteriorcingulate
60: ctx-rh-rostralmiddlefrontal
61: ctx-rh-superiorfrontal
62: ctx-rh-superiorparietal
63: ctx-rh-superiortemporal
64: ctx-rh-supramarginal

Figure 8: Chord diagram of reduced connectivity in low-reading group.

The most discriminative edges are concentrated in cross-hemispheric connections linking frontal and parietal lobes, specifically between superior and middle frontal gyri (including BA46), the superior parietal lobule, and the precuneus. These findings underscore the role of distributed fronto-parietal networks in supporting reading performance.

In simple terms, individuals with higher reading ability show "stronger bridges" between the left and right hemispheres, enabling more efficient information transfer. By contrast, low-reading individuals exhibit weaker cross-hemisphere integration. This highlights structural connectivity as a neural substrate of individual differences in reading ability.

# N  LARGE LANGUAGE MODELS USAGE STATEMENT

Large Language Models were only used to aid or polish writing.

