# OpenReview forum: "GP-STPCA: Generalized Power Method for Sparse Tensor Principal Component Analysis"
_ICLR.cc/2026/Conference — ICLR 2026 Conference Withdrawn Submission_

### Official Review · Reviewer_p57b · 2025-10-25

**Soundness:** 2
**Presentation:** 2
**Contribution:** 1
**Rating:** 2
**Confidence:** 4

**Summary:**

A method for Tucker-based sparse tensor PCA is proposed. The authors consider a unified setting for both l0- and l1-constraints penalties and single-unit or block formulations. Experiments are conducted on synthetic data, several images, and brain connectome data.

**Strengths:**

1. A new method for Tucker-based sparse PCA is proposed.

2. There is some theoretical analysis on it.

**Weaknesses:**

1. The core modeling insights are insufficient. It seems that the authors straightforwardly extend (1)-based sparse PCA methods and (3)-based Tucker sparse PCA formulation. Formulation (4) considers unfolding which is a standard operation in Tucker-based methods. Thus, the modeling novelty is limited.

2. Due to the high technical similarity of the processing of $\ell_0$, $\ell_1$ both in element-wise and block-wise sparsity, the so-called unified modeling is of little essential significance.

3. The theory analysis under the Assumptions 1-3 is simple, and of little theoretical significance.

4. The experiments are insufficient. On synthetic data, the experiments only use rank-1 tensors, which is a very special and unrealistic case. The dataset sizes are also limited. Additionally, the authors only compare against GP-STPCA.

**Questions:**

See the above weaknesses.

---

### Official Review · Reviewer_brfT · 2025-10-29

**Soundness:** 2
**Presentation:** 1
**Contribution:** 1
**Rating:** 0
**Confidence:** 5

**Summary:**

This paper extends sparse Principal Component Analysis (PCA) from the matrix setting to higher-order tensor data under the Tucker decomposition framework. The authors formulate the Sparse Tucker PCA (STPCA) problem as a multi-mode sparse low-rank approximation and propose an alternating optimization algorithm based on the generalized power method (GPower).

**Strengths:**

The paper extends sparse Principal Component Analysis (PCA) to a multi-mode Tucker structure, aiming to develop a theoretically grounded and computationally tractable framework for sparse tensor principal component analysis (STPCA).

**Weaknesses:**

1. However, similar ideas have already been explored in prior works such as Sparse Higher-Order PCA [1] and sparseGeoHOPCA. Therefore, the novelty of the proposed framework requires clearer justification and differentiation from these existing approaches.

2. The provided convergence guarantee only applies to the simplified subproblems, not to the full alternating multi-block optimization of the entire STPCA.

3. The motivation and distinction from prior tensor sparse PCA works are not clearly articulated. Similar ideas already exist, such as Sparse Higher-Order PCA (Sparse HOPCA) by Allen (2012) and sparseGeoHOPCA, which also impose sparsity on Tucker/CP decompositions. The paper needs to explicitly discuss these connections and clarify what is genuinely new beyond previous formulations.

4. The experiments are underdeveloped. It compares against only one baseline method, and the real-data evaluation consists of just six color images for quantitative comparison. As it stands, the empirical section is insufficient to validate the proposed method.

5. Writing and Presentation Issues:

a. The introduction lacks a clear problem statement and motivation. For example, the sentence “However, introducing sparsity makes TPCA a non-convex and generally NP-hard problem” is abrupt and undefined—what does “TPCA” specifically refer to, and where is it cited?

b. Tucker decomposition, a central concept, is never introduced in the Introduction, which makes it hard for readers to understand the context.

c. Figure 1 (e.g., the black dots in the figure) is confusing—they could be misinterpreted as indicating sparse noise rather than sparsity in the factors.

d. Using “In this paper, ...” to start the Introduction is very uncommon in academic writing.

e. Equations such as (1) and (3) are presented without adequate intuition.

6. The related work section relies heavily on older literature. The paper fails to discuss recent developments in sparse tensor decomposition, low-rank recovery, or structured regularization over the last five years.

[1] Sparse higher-order principal components analysis,Genevera I. Allen

**Questions:**

See the weaknesses 3-6.

---

### Official Review · Reviewer_EtrH · 2025-10-30

**Soundness:** 3
**Presentation:** 2
**Contribution:** 2
**Rating:** 2
**Confidence:** 3

**Summary:**

This paper proposes GP-STPCA, a generalized power–based framework for efficient and interpretable sparse tensor PCA. It achieves superior accuracy and efficiency over prior methods through principled reformulation and theoretical convergence guarantees.

**Strengths:**

Strength
1. Provides solid theoretical grounding by establishing equivalence to the original sparse objective and proving step-size convergence under mild convexity assumptions.
2. Achieves notable computational efficiency by reducing the search space via mode-wise convex reformulation, yielding up to 50–100× speedups over baselines.
3. Produces well-localized and physically meaningful sparse factors across tensor modes, enhancing interpretability.

**Weaknesses:**

Weakness
1. Experimental evaluation is limited — the image reconstruction results are based on only a few examples and do not convincingly demonstrate general performance.
2. The algorithm may be sensitive to initialization, risking convergence to local optima.
3. Several sparsity-related hyperparameters (γₙ, μₙ, kₙ) require manual tuning with little guidance.
4. Theoretical guarantees depend on strong convexity assumptions that may not hold in real-world non-convex problems.

**Questions:**

Questions
1. Larger-scale image reconstruction experiments are needed — evaluating only a few images is insufficient to assess overall model capability.
2. Can the framework be extended to handle dynamic or variable-order tensors, such as video or time-evolving data?
3. Could GP-STPCA be integrated with nonlinear tensor models or deep architectures (e.g., autoencoders, deep factorization networks)?
4. How robust is the method to noise and corruption? Would robust penalties (e.g., Huber loss, correntropy) improve stability?

**Details Of Ethics Concerns:**

No ethics concers.

---

### Official Review · Reviewer_LpQ7 · 2025-11-03

**Soundness:** 2
**Presentation:** 2
**Contribution:** 2
**Rating:** 2
**Confidence:** 4

**Summary:**

The paper proposes GP-STPCA, a Tucker-based framework that reduces sparse tensor PCA (STPCA) to a sequence of sparse matrix PCA subproblems that are allegedly solvable efficiently with the generalized power method.The authors handle both ℓ0 and ℓ1penalties, in single-unit and block formulations, add a “pattern-finding + post-processing” procedure, and prove a stepsize convergence bound for a generalized gradient scheme under strong convexity assumptions.

**Strengths:**

1.Reduction to right-sparse matrix PCA is sensible for column-dominant unfoldings; writing the mode-wise objective in maximization form and using power-like updates is well motivated.
2.The post-processing step (polar/SVD refinement on the identified support) is a practical fix to value distortion caused by ℓ1shrinkage.

**Weaknesses:**

1.Questionable novelty relative to established sparse PCA with generalized power. The main technical engine (power-type thresholding on a reduced problem) is lifted directly from Journee et al. (2010). The paper reframes STPCA as a stack of such subproblems but gives little truly new algorithmic substance beyond bookkeeping, deflation, and a routine polar normalization. The “equivalence” and “unified view” read more like repackaging than a substantive advance.
2.Missing baselines include modern sparse-Tucker/ALS variants, convex relaxations, and other iterative hard-thresholding approaches on tensor modes. The chosen baseline (sparseGeoHOPCA) is a single family; the study lacks breadth.
3.For the experimental part, it is suggested that the authors add more data sets and tensor-based algorithms . The authors did not carry out more relevant experiments well to support the research motivation of this paper.

**Questions:**

1. The number of experimental samples in the main text of the paper is relatively small. For natural high-order representation structures like tensors, it would be more convincing to use video datasets. Why didn't the author incorporate the above-mentioned high-dimensional data structures?

2. Regarding the approach proposed by the author, is it possible to conduct a theoretical analysis of the error and reach such an error bound?

**Details Of Ethics Concerns:**

As mentioned above.

---

### Note · Authors · 2025-12-03

I have read and agree with the venue's withdrawal policy on behalf of myself and my co-authors.